# Coastal terrestrial emissions modify the composition and optical properties of aerosols in marginal seas

Kuanyun Hu[1], Narcisse Tsona Tchinda[1], Kun Li[1], Hartmut Herrmann[2,3], Jianlong Li[1,*], Lin Du[1,3,4,*]

[1]Qingdao Key Laboratory for Prevention and Control of Atmospheric Pollution in Coastal Cities, Environment Research Institute, Shandong University, Qingdao 266237, China
[2]Atmospheric Chemistry Department (ACD), Leibniz Institute for Tropospheric Research (TROPOS), Leipzig 04318, Germany
[3]School of Environmental Science and Engineering, Shandong University, Qingdao 266237, China
[4]State Key Laboratory of Microbial Technology, Shandong University, Qingdao 266237, China.

*Correspondence to*: Lin Du (lindu@sdu.edu.cn); Jianlong Li (jianlongli@sdu.edu.cn)

**Abstract.** Aerosols originating from marginal seas significantly contribute to regional and global air pollution burdens and climate regulation. However, the responses of marine aerosols to terrestrial transport remain uncertain. In this work, we investigated chemical composition and properties of aerosols over Bohai Sea (BS) and Yellow Sea (YS). Our observation results indicate that in the BS and YS regions in summer, the proportion of terrestrial boundary layer air masses is lower than that of marine boundary layer air masses. But the terrestrial characteristics of aerosols in these two regions are still apparent. The characteristics of carbon component ratios (mean: 5.58–12.11 for OC/EC, 0.48–0.58 for WSOC/OC) in aerosols are similar to those in coastal cities, and the proportion of non–sea–salt ions (> 80%) is significantly higher than that of sea salt ions. Humic–like substances (HULIS) and water–insoluble organic compounds (WISOC) mainly contain anthropogenic components (oxygenated aromatic compounds, nitrogen-containing aromatic compounds, anthropogenic surfactants, fossil fuel-derived SOA). The $\delta^{13}C_{TC}$ characteristics and positive matrix factorization model (PMF) reveal that biomass burning contributes 60–80% to carbonaceous species and marine primary emissions contribute 20% to aerosols. According to the height of air mass trajectories and the half–life (15–20 h) of light–absorbing components, we determined the coastal region within 190–260 km of the coastline and the coastal waters as key regions controlling the composition of BS and YS aerosols. Our results emphasize that during summer when the influence of marine air masses increases, the terrestrial characteristics of BS and YS aerosols remain evident, being related to air mass transport from coastal terrestrial regions.

**Keywords:** Marginal Seas; Terrestrial emission; Source apportionment; Backward trajectories; Molecular composition

# 1 Introduction

Marine aerosols constitute one of the largest natural sources of aerosols globally, with estimated emissions of ~5000 Tg yr⁻¹ originating from sea spray alone. They contribute to the global aerosol load, leading to strong influences on global climate (Russell et al., 2023; Quinn et al., 2015). Unlike remote oceans, marginal seas are critical sites for land-sea interactions. As river and atmospheric deposition deliver substantial nutrients and anthropogenic pollutants into seawater, they promote emissions of marine biogenic substances and facilitate the aerosolization of anthropogenic pollutants during sea spray production, thereby reintroducing them into the atmosphere (Park et al., 2019; Zhang et al., 2023a; Franklin et al., 2022). Consequently, the chemical composition and properties of the atmosphere over marginal seas are co-regulated by both terrestrial transport and marine emission (Huang et al., 2022; Mo et al., 2022; Zhong et al., 2024), resulting in aerosols that are more complex than those over remote oceans. This complexity causes significant challenges for source apportionment and impact assessment of marginal sea aerosols, particularly in regions strongly influenced by human activities.

Marginal seas, located at transitional zones between continents and open oceans, exhibit unique aerosol characteristics blending terrestrial and marine sources. Unlike open oceans, marginal sea aerosols are frequently influenced by continental air mass outflows carrying both anthropogenic and natural emissions (Zhou et al., 2023; Liu et al., 2022). Major rivers also discharge anthropogenic pollutants and freshwater into marginal seas, causing marginal seawater pollution and salinity changes (Chen et al., 2024; Park et al., 2022). Besides, this process also introduces nutrients (e.g., nitrogen, phosphorus, iron) that significantly affect surface marine ecosystems and biogeochemical cycles. Subsequent changes in seawater properties further modify air–sea exchange processes (e.g., sea spray emission, marine volatile organic compounds (VOCs) emissions) (Zheng and Zhai, 2023; Zhang et al., 2023b). Marginal seas are adjacent to coastal terrestrial regions with important human activities and high anthropogenic emissions. Abundant terrestrial gaseous precursors and aerosols undergo atmospheric transformation during their transport to the ocean, altering the composition of marine aerosols (Hu et al., 2022; Zhao et al., 2024). These different sources, transport and transformation processes enhance the complexity of marginal sea aerosols, making composition and source analysis crucial for understanding the land–sea interactions.

Geographically, Bohai Sea (BS) and Yellow Sea (YS) are among the largest marginal sea systems in the world (Yu et al., 2023). BS and the northern YS are connected through the narrow Bohai Strait and are predominantly surrounded by land, while the southern YS is linked to the western Pacific Ocean (Zheng and Zhai, 2023). These regions are characterized by high marine productivity, significant terrestrial transport and riverine inputs. Consequently, land–sea interactions exert a pronounced influence on the composition and properties of aerosols over these seas, making them ideal sites for studying the impact of land–sea interactions on marine aerosols. For instance, recent studies have shown that some inorganic components exhibit latitudinal gradient characteristics in the BS and the YS. In the BS and the northern YS heavily polluted by terrestrial sources, the formation pathway of atmospheric $NO_3^-$ is dominated by anthropogenic hydrocarbon, which is significantly different from that in the southern YS, weakly influenced by terrestrial activities (Zhao et al., 2024). Due to the significant contributions of coal combustion and biomass burning, chlorine enrichment occurs in the aerosols of the BS, while chlorine

depletion due to aerosol aging is observed in the aerosols of the southern YS (Liu et al., 2022). However, these studies mainly focused on spring and winter emissions, which are strongly influenced by terrestrial transport. In summer, due to the influence of monsoon, the proportion of marine air masses reaching BS and YS is higher. However, the differences in the composition and properties of marine aerosols among different sea regions are yet to be explored.

To elucidate the characteristics of aerosols over the BS and YS in summer, total suspended particulate (TSP) and PM$_{2.5}$ samples were collected from July to August 2023. This study analyzes the characteristics of air masses, aerosol sources, and chemical composition to understand the chemical characteristics of aerosols over the BS and YS in summer.

## 2 Materials and Methods

### 2.1 Sample collection and chemical analysis

The cruise campaign was conducted aboard the R/V Lanhai 101, with samples taken in the BS and northern YS from July 15 to 23, 2023, and in the southern YS from August 11 to 13, 2023. Two high–volume air samplers (Laoying 2031) were located on the first deck of the ship to collect TSP and PM$_{2.5}$ samples at a flow rate of 1.05 m$^3$ min$^{-1}$ on quartz filters (preheated at 500°C for 4 hours). In order to avoid the impact of ship exhaust emissions and ensure that the collected airflow comes from the bow of the ship, the sampler only starts sampling when the ship is sailing. The sampling sites are shown in Fig. 1A. The collection times for each TSP and PM$_{2.5}$ samples were approximately 12 h and 24 h, respectively. The concentration of particles was calculated by dividing the mass of particles by the volume of collected air.

The basic experimental flowchart and detailed experimental procedures are shown in Text S1 and Fig. S1 in the supplement. Briefly, the experimental workflow involved quantifying OC and EC using a semi–continuous OC/EC analyzer, with instrument calibration via sucrose standards. Filter samples underwent aqueous ultrasonic extraction, followed by filtration and measurement for WSOC analysis using a TOC analyzer, ensuring analytical precision through triplicate measurements with < 3% relative standard deviation. Solid–phase extraction was used to isolate hydrophilic WSOC (HPWSOC) and HULIS. The recovery rate of HULIS was above 90%, and they were concentrated and redissolved for carbon quantification. Water–soluble ions (Na$^+$, NH$_4^+$, K$^+$, Mg$^{2+}$, Ca$^{2+}$, Cl$^-$, SO$_4^{2-}$, and NO$_3^-$) were extracted from filters and analyzed by ion chromatography. In order to verify the stability of the instrument, a standard solution of 5 mg L$^{-1}$ of the above-mentioned ions was used to calibrate the instrument after every 5 samples to ensure that the relative standard deviation of repeated measurements of the same sample is less than 6%.

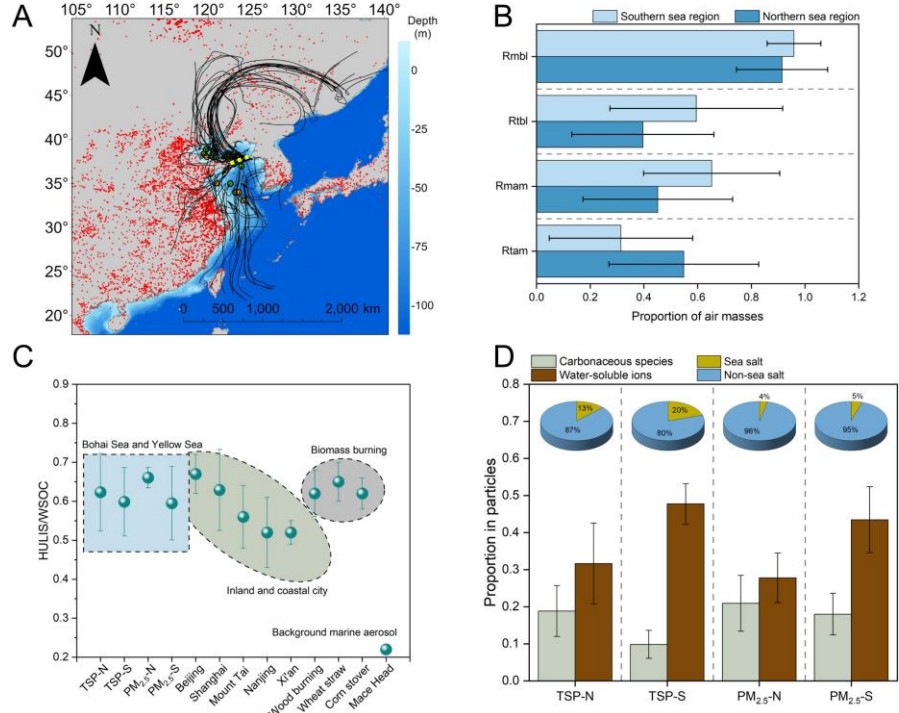

**Figure 1. (A)** Trajectories of air masses arriving at the Yellow Sea and Bohai Sea during the sampling period. The simulated air mass transport time is 72 hours. Green, yellow and red points represent TSP, PM$_{2.5}$ samples and fire points, respectively. Fire point information comes from https://firms.modaps.eosdis.nasa.gov/map. The yellow dashed line represents the boundary between the northern and southern sea regions. **(B)** Retention ratio of air masses over land and ocean, as well as the retention ratio of boundary layer air masses. Rmbl stands for Retention ratio of marine boundary layer air masses, Rtbl stands for Retention ratio of terrestrial boundary layer air masses, Rmam stands for Retention ratio of marine air masses and Rtam stands for Retention ratio of terrestrial air masses. **(C)** Differences in HULIS/WSOC ratio at different sampling points. **(D)** Proportion of carbonaceous species and water–soluble ions in particles. The pie charts represent the proportion of sea salt ions and non–sea salt ions in the total ions of each sea region. N and S indicate the northern and southern sea regions, respectively.

## 2.2 Optical properties measurement

The light absorption properties of MSOC (extracted with methanol), WSOC, HULIS and HPWSOC were measured using a UV–vis spectrometer (P9, Shanghai Mapada, China). The light absorption of extracts was measured in the range of 190–800 nm with an interval of 1 nm. A fluorescence spectrometer (Duetta™, Horiba Scientific, Japan) was used to measure the EEM spectra of extracts. Both the excitation and emission wavelength ranges of EEM were 250–600 nm, with 5 and 2 nm intervals, respectively.

## 2.3 High resolution mass spectrometry analysis and data processing

The molecular characterization of HULIS and WISOC was conducted by a liquid chromatography–mass spectrometry system consisting of an Ultra–High Performance Liquid Chromatography (UltiMate 3000, Thermo Scientific, USA) and Quadrupole–Time of Flight Mass Spectrometer (Q–TOF MS, Bruker Impact HD, Germany). This spectrometer is equipped with an electrospray ionization (ESI) source and molecules were measured in the negative mode (ESI (–)). The data were processed using the Bruker Compass DataAnalysis 4.2 software. Based on the maximum number of atoms, the identified formulas are limited to $^{12}C_{70}{}^{1}H_{140}{}^{16}O_{25}{}^{14}N_3{}^{32}S_1$ and mass tolerance was set at ± 5 ppm. In order to eliminate unreasonable formulas, H/C, O/C, N/C and S/C were set within the ranges of 0.3–3, 0–3, 0–0.5 and 0–0.2, respectively. For the $C_cH_hO_oN_nS_s$ formula, the double bond equivalents (DBE) were calculated using the following equation:

$$DBE = \frac{(2c+2-h+n)}{2} \tag{1}$$

The modified aromaticity index (AI) indicates the degree of unsaturation of the molecular formula (the number of carbon double bonds and rings). AI can be calculated using the following equation:

$$AI = \frac{1+c-s-0.5\times(o+h)}{c-0.5\times o-n-s} \tag{2}$$

The oxidation state of the carbon atom ($OS_C$) can be defined as the charge it would acquire from bonds involving more electronegative atoms, while losing charge from bonds involving less electronegative atoms. $OS_c$ is usually used to evaluate the degree of oxidation of organic compounds in the atmosphere and it can be calculated as:

$$OS_c = 2\times\frac{O}{C} - \frac{H}{C} \tag{3}$$

In order to consider the intensity of each formula and its contribution to the overall property, the intensity-weighted parameters ($X_w$) are calculated using the following equation:

$$X_w = \frac{\sum(Int_i\times x_i)}{\sum Int_i} \tag{4}$$

where $x_i$ represents the calculated molecular parameters (O/C, H/C, N/C, S/C, DBE, AI, $OS_c$) for formula $i$, $Int_i$ is the intensity of formula $i$.

## 2.4 Retention ratio of air masses

The retention ratios of terrestrial air masses (Rtam) and marine air masses (Rmam) are calculated and modified from the method proposed by previous studies (Zhou et al., 2023; Xu et al., 2025):

$$Rtam = \sum_{m=1}^{M}\left(f\times\frac{\sum_{i=1}^{N_{land}} e^{-\frac{t_i}{72}}}{\sum_{i=1}^{N_{total}} e^{-\frac{t_i}{72}}}\right) \tag{5}$$

$$Rmam = \sum_{m=1}^{M} (f \times \frac{\sum_{i=1}^{N_{ocean}} e^{-\frac{t_i}{72}}}{\sum_{i=1}^{N_{total}} e^{-\frac{t_i}{72}}}) = 1 - Rtam \tag{6}$$

where $N_{land}$ and $N_{marine}$ are total number of trajectory endpoints located over the land and the ocean, respectively, $f$ is the proportion of each air mass trajectory at the same latitude and longitude point, $M$ is the number of trajectories of air masses reaching the same latitude and longitude point, $N_{total}$ is the total number of trajectory endpoints, $t_i$ is the backward tracking time (in hours) of the $i$–th endpoint. $e^{-\frac{t_i}{72}}$ is the weighting factor related to the tracking time as the diffusion of air mass and deposition of particles take place along the transport. Hence, the regions corresponding to longer backward tracking time have a weaker influence on the receptor site than the nearby regions.

Considering that pollutants mainly accumulate in the boundary layer, in order to further evaluate the impact of pollutants in the boundary layer on the marine atmospheric environment, the retention ratio of air masses within the boundary layer was calculated as:

$$Rtbl = \sum_{m=1}^{M} (f \times \frac{\sum_{i=1}^{N_{below}} e^{-\frac{t_i}{72}}}{\sum_{i=1}^{N_{land}} e^{-\frac{t_i}{72}}}) \tag{7}$$

$$Rmbl = \sum_{m=1}^{M} (f \times \frac{\sum_{i=1}^{N_{below}} e^{-\frac{t_i}{72}}}{\sum_{i=1}^{N_{ocean}} e^{-\frac{t_i}{72}}}) \tag{8}$$

where $N_{below}$ is the number of marine or land endpoints with the heights below the boundary layer. Rtbl indicates the retention ratio of terrestrial boundary layer air mass. Rmbl indicates the retention ratio of marine boundary layer air mass. Large Rtbl or Rmbl values indicate that the movements of air over the ocean are mainly confined within the boundary layer.

**2.5 Model analysis**

Part of every filter (10 cm$^2$) was pretreated with HCl vapor to remove inorganic carbon. The pretreated samples were then wrapped in tin cups for stable carbon isotope analysis of total carbon ($\delta^{13}$C) using an isotope ratio mass spectrometer (253 plus, Thermo Fisher Scientific, Germany). The Bayesian mixing model can be used to characterize sources of carbonaceous aerosols and quantify their contribution. Besides, the positive matrix factorization model (PMF) was also used to evaluate the contribution of potential sources to marine aerosols. Air mass trajectory was analysed using the Hybrid Single–Particle Lagrangian Integrated Trajectory model. Given that the average atmospheric lifetime of organic and inorganic substances in aerosols reported in literature is mainly around 3–6 days (Pai et al., 2020; Liu et al., 2012), and referring to the commonly used air mass simulation time (Cohen et al., 2015), we chose to simulate the trajectory of the air mass within 72 hours. The minimum R squared method (MRS) was used to model the concentration of secondary organic carbon (Wu et al., 2019a). Parallel factor analysis was used to analyze fluorescent components in organic compounds, and the analysis procedure follows that from a previous study (Stedmon and Bro, 2008). The calculation and description details of above models are shown in Text S2–S5.

## 3. Results

### 3.1 Air mass characteristics in investigated regions

Significant differences were observed in the origins of air masses over the BS and YS during the sampling period. As shown in Fig. 1A, air masses over the BS and northern YS originated from Northeast China and Northeast Asia, and were classified as terrestrial air masses, while air masses over the southern YS originated from South China and the ocean, and were classified as terrestrial-marine mixed air masses. Given the differences in the origins of air masses, the BS and northern YS were categorized as the northern sea region, and the southern YS as the southern sea region. Furthermore, when air masses are transported over different underlying surfaces, the substances carried to the receptor point are also different. Consequently, we calculated the retention ratio of air masses over land and sea based on trajectory paths and transport times to assess the degree of terrestrial and marine influence on aerosols over these two sea regions (Xu et al., 2025). Results show that the retention ratio of terrestrial air mass (Rtam) in the northern sea region (0.55 ± 0.28) was approximately twice that in the southern sea region (0.31 ± 0.27) (Fig. 1B), and the retention ratio of marine air masses (Rmam) in the southern sea region was higher than Rtam. This suggests that aerosols over the northern sea region appeared to be more influenced by terrestrial sources, while those in the southern sea region exhibited characteristics of marine emissions. However, as atmospheric pollutants are predominantly concentrated within the boundary layer, it is necessary to consider the impact of variations in air mass transport height on marginal sea aerosols.

We further calculated the retention ratio of terrestrial and marine boundary layer air masses, and found apparent differences in the transport heights of air masses reaching the northern and southern sea region. Part of air masses originating at relatively high altitudes, up to 3000 m, and that arrived at the northern sea region, only descended significantly near the sampling region. In contrast, air masses reaching the southern sea region remained at low altitudes throughout the transport (Fig. 2A). Based on previous studies, the atmospheric boundary layer height typically ranges from several hundred meters to 1500 meters, often remaining below 1000 meters (Li et al., 2021; Zhang et al., 2020). The satellite observation results also showed that the boundary layer height of the study region during the sampling period was mainly below 1000 m or 500 m, and the difference in retention ratio of air masses calculated at these boundary layer heights was not significant (Table S1 and Fig. S2). Therefore, we chose 1000 m (equivalent to ~900 hPa) as the upper limit of the boundary layer (Deng et al., 2022). We calculated the retention ratio of terrestrial boundary layer air mass (Rtbl) and retention ratio of marine boundary layer air mass (Rmbl), both reflecting the residence time of air masses within the boundary layer and serving as measures of the influence of terrestrial and marine boundary layer pollutants on the sampling region. Fig. 1B shows that Rtbl in the northern sea region (0.44 ± 0.26) is lower than that in the southern sea region (0.60 ± 0.32). This indicates that although most of air masses reaching the northern sea region originate from land, they rarely move within the boundary layer or have a short transport time over land. Overall, air masses reaching BS and YS are characterized by high Rmbl.

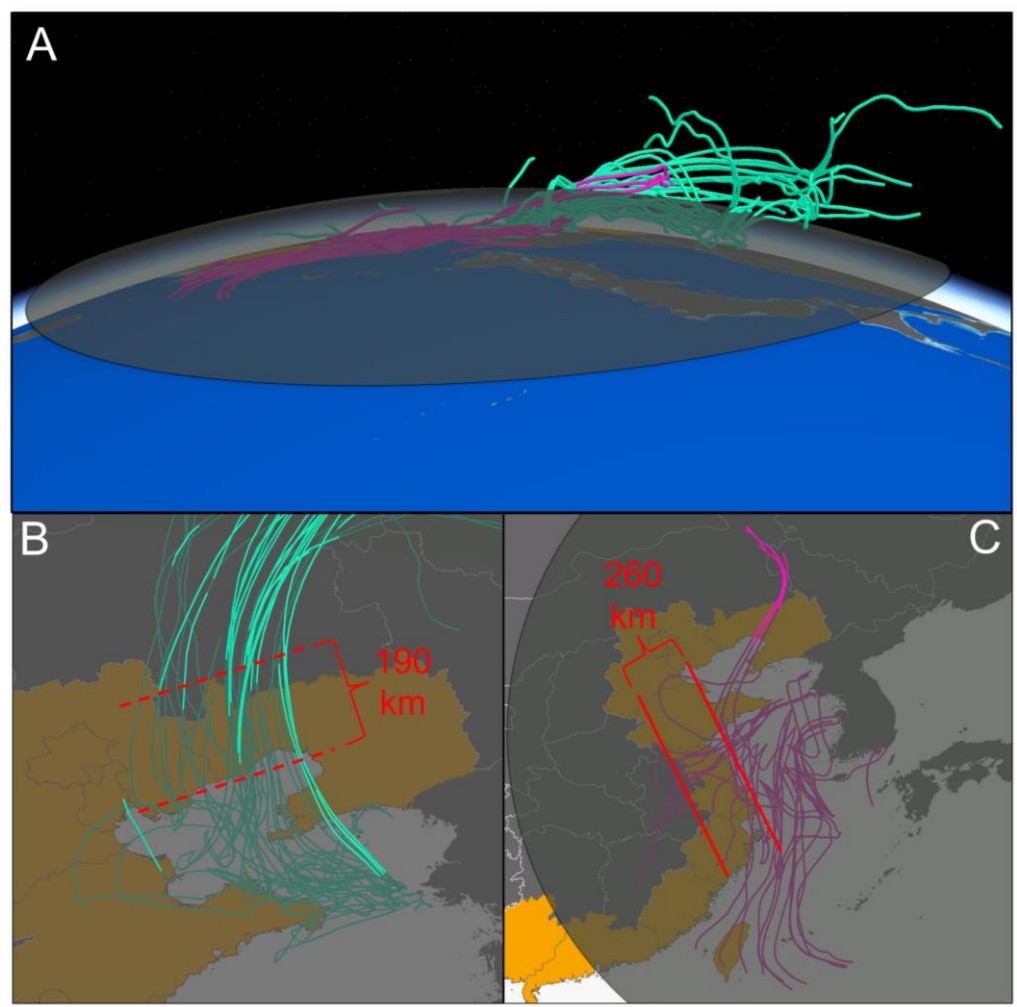

**Figure 2. (A) Changes in trajectory height during the transport of air masses. (B) Top view of the air mass reaching the northern sea region. (C) Top view of the air mass reaching the southern sea region. The gray circular surface in Panel A (at an altitude of 1000 m), and brighter trajectory colors indicate the transport of air masses above the boundary layer. The orange region represents the coastal terrestrial regions of China. The distance in the figure is calculated using ArcGIS Pro software.**

### 3.2 Characteristics of summer aerosol composition

During the sampling period, we observed that the retention ratio of marine boundary layer air masses (Rmbl) in the two sea regions (> 90%) was higher than that of terrestrial boundary layer air masses (Rtbl) (< 60%) (Fig. 1B). Considering that atmospheric pollutants mainly constrained within the boundary layer, high Rmbl theoretically indicates that aerosols are more affected by marine emissions (Zhou et al., 2023; Zhou et al., 2021; Yan et al., 2024). However, our results clearly showed terrestrial characteristics in aerosols. Firstly, the concentration of particulate matter is significantly correlated with the calculated Rtbl ($R = 0.829$ and $P < 0.01$ for TSP, and $R = 0.811$ and $P < 0.05$ for PM$_{2.5}$), indicating that aerosols in marginal seas are greatly influenced by terrestrial sources (Fig. S3A and B). Secondly, the concentration ratio of organic carbon (OC)

to elemental carbon (EC) (OC/EC) in this study was nearly ten times higher than that in aerosols over the remote northwest Pacific Ocean, which are significantly influenced by marine air masses (Boreddy et al., 2018). The OC/EC ratio ($6.53 \pm 2.39$ for TSP, $9.29 \pm 4.42$ for $PM_{2.5}$) was similar to ratios for coal combustion and biomass burning aerosols (2.5–12.7) and higher than those for mobile source aerosols (1.0–4.2) (Wang et al., 2022). The ratio of water-soluble organic carbon (WSOC) to OC (WSOC/OC) was approximately 50–90%, similar to results from previous studies in inland, coastal cities, and marginal seas (Table S2). This reflects the apparent terrestrial characteristics of the carbonaceous components in BS and YS summer aerosols. The ratio of humic-like substances (HULIS) to WSOC (HULIS/WSOC) ($0.63 \pm 0.16$ for TSP, $0.66 \pm 0.12$ for $PM_{2.5}$) was comparable to values in coastal cities and biomass burning samples, being significantly higher than in background marine aerosols (Fig. 1C). Under the assumption that all $Na^+$ originates from the sea, non-sea-salt (nss) ions accounted for over 80% of total ions. However, a previous report has suggested that 50–80% of $Na^+$ in coastal atmospheric aerosols may originate from anthropogenic sources (Wu et al., 2024), implying that the actual sea-salt ion contribution might even be lower (Table S3).

Regionally, aerosols over the northern sea region exhibited high OC/EC (see Table S2). Contrary to the OC/EC, the WSOC/OC ratio was high in the southern sea region. Although higher OC/EC and WSOC/OC may indicate the influence of secondary organic aerosol (SOA), the contrary ratio characteristics in different sea regions indicate that SOA formation is not enough to explain this phenomenon. For example, we found that OC exhibit strong correlations with nss–$Ca^{2+}$ and EC, while WSOC exhibit not only a moderate correlation with secondary inorganic ions ($NH_4^+$, $NO_3^-$ and nss–$SO_4^{2-}$), but also a positive correlation with nss-$K^+$ and EC (Fig. S3A and B). This indicates that in addition to secondary transformation, primary combustion and non–combustion sources contribute to OC and WSOC (Cai et al., 2020). Besides, the SOC calculated using the minimum $R^2$ method exhibits a moderate correlation with WSOC, with a slope lower than 1 (Fig. S3C). This indicates that SOC may partially be water insoluble (Zhang et al., 2022b). The majority of OC/EC and WSOC/OC ratios in field studies fall between 5–10 and 0.4–0.7, respectively (Nayak et al., 2022; Arun et al., 2021; Wu et al., 2019b; Zhang et al., 2022a; Chen et al., 2020; Chen et al., 2023; Luo et al., 2020; Rajeev et al., 2022; Zhao et al., 2023) (see Fig. S3C), consistent with the results of this study. In addition, no similar changes were observed between the trends of OC/EC ratios and WSOC/OC ratios, indicating that the generation of SOA may not lead to a simultaneous increase in OC/EC ratios and WSOC/OC ratios. Therefore, combustion sources, non–combustion sources, and atmospheric transformation may be the reasons for the difference of ratios between the two sea regions. But the EC concentration in the southern sea region is high, indicating that the impact of the primary combustion source seems to be greater (Fig. S3D). Additionally, in the southern region, particularly in TSP samples, sea-salt ions constituted a larger proportion of all ions, indicating stronger primary marine emissions. For example, high wind speeds at 10 m above the sea surface during sampling in the southern sea region (Fig. S4) enhanced the mechanical processes at the marine surface, thereby increasing emissions of marine source ions. Overall, non–sea–salt ions still account for the largest proportion (> 80%) of all ions, indicating a low contribution of marine sources to ions composition.

High Rmbl does not indicate significant marine characteristics of BS and YS summer aerosols. In contrast, the apparent terrestrial characteristics of aerosols may have connections with strong coastal terrestrial emissions and weaker marine emissions in summer. Overall, the terrestrial air masses in the BS and YS during summer mainly transport pollutants in the

boundary layer of Bohai Rim region (including Liaoning province, Shandong province, as well as Beijing–Tianjin–Hebei region) and Jiangsu Province (Fig. 2B and C). The Bohai Rim region is one of the regions with the highest pollutant emissions in China. OC and EC emission inventories indicate that fossil fuel combustion alone in China emitted 2.1 Tg OC and 1.5 Tg EC in 2015. Even with implemented emission reduction measures, projected annual OC and EC emissions for 2030 remain above 1 Tg, and $NO_x$ and $SO_2$ emissions are close to 20 Tg annually (Lu et al., 2020). Despite covering only ~5% of China's land area, the Bohai Rim region is the region with the highest carbon emission intensity in China, contributing over 20% of China's total OC and EC emissions (Fang et al., 2018). The BS and YS are adjacent to the Bohai Rim region and are situated on their air mass transport paths, thus significantly influenced by the outflow of terrestrial air masses from this region. Furthermore, sea surface chlorophyll–a (Chl–a) concentration is a key indicator of marine phytoplankton biomass/biogenic emissions. We found that summer is not the period of maximum phytoplankton biomass in the BS and YS (Fig. S5). Spatially, except for the BS and region near the Yangtze River estuary, the surface Chl–a concentration in the sea region where the marine boundary layer air mass passes through is not very high. Although some trace bioactive gases are emitted by marine organisms, they are much smaller in magnitude than the transported terrestrial components.

### 3.3 Molecular characteristics and recognition of typical compounds

The molecular composition of aerosols shows distinct terrestrial characteristics of BS and YS summer aerosols (Table S4). The intensity-weighted double bond equivalent ($DBE_w$) (5–9) and aromaticity index ($AI_w$) (0.2–0.4) of organic compounds in this study were higher than values typical of marine air masses ($DBE_w$: 3.91, $AI_w$: 0.15) and were closer to values influenced by terrestrial air masses ($DBE_w$: 5–7, $AI_w$: 0.20–0.30) (Mo et al., 2022), indicating that compounds in BS and YS summer aerosols exhibit stronger terrestrial characteristics. Furthermore, although Rtbl in the northern sea area is lower than that in the southern sea region, the $DBE_w$ and $AI_w$ of HULIS and water-insoluble organic carbon (WISOC) in its aerosols are higher than those in the southern sea region.

Compounds with relatively high intensity in typical chromatographic peaks can reflect potential aerosol sources. Among peaks identified with a signal-to-noise ratio (S/N) ≥ 5, the characteristic molecules were predominantly anthropogenic SOA. In HULIS, characteristic molecules detected in samples from both sea regions were aromatic and unsaturated SOA, including oxygenated aromatic compounds, nitrogen-containing aromatic compounds, and fossil-fuel-derived SOA. Oxygenated aromatic compounds are characterized by high AI (≥ 0.5) and DBE (≥ 5) (Fig. S6). Among these, chromatographic peaks corresponding to phthalic acid ($C_8H_5O_4^-$) and benzoic acid ($C_7H_5O_2^-$) exhibited high S/N. These compounds typically originate from biomass burning, fossil fuel combustion, and atmospheric oxidation of toluene (Santos et al., 2019; Boreddy et al., 2017). Nitrogen–containing aromatic compounds were primarily nitroaromatic compounds (NACs), with typical molecules including $C_6H_4NO_3^-$, $C_6H_3N_2O_5^-$, $C_7H_4NO_5^-$, $C_7H_6NO_3^-$, and $C_7H_3N_2O_7^-$ (Fig. S7A and C). These NACs are widely observed in urban and coastal environments and typically arise from biomass burning, coal combustion, vehicular emissions, or photochemical reactions involving volatile organic compounds (VOCs) containing benzene rings (e.g., cresols, catechol, methylcatechol)

(Wang et al., 2019; Jiang et al., 2024). Highly aromatic and unsaturated CHOS compounds, typically emitted directly from fossil fuel sources, were also detected in HULIS from northern sea region aerosols (Fig. S8A).

In addition to these aromatic molecules, some unsaturated molecules were detected in HULIS. These included biogenic SOA (α-pinene, β-pinene, α-terpinene, and terpinolene SOA), diesel-derived SOA, and unsaturated fatty acid substances. Detected biogenic SOA molecules were primarily $C_{10}H_{14}NO_7S^-$, $C_{10}H_{16}NO_7S^-$, $C_{10}H_{16}NO_8S^-$, $C_{10}H_{17}N_2O_{11}S^-$, and $C_9H_{15}O_7S^-$ (Fig. S9).

Among these, the peak corresponding to $C_{10}H_{16}NO_7S^-$ had the highest S/N. This compound is also a major product derived from biogenic monoterpenes in smog chamber simulations (Wang et al., 2021; Surratt et al., 2008; Kuang et al., 2016). The precursors of these biogenic SOA are most likely isoprene and monoterpenes released by terrestrial plants in coastal terrestrial regions and phytoplankton in coastal waters. Although marine isoprene and monoterpenes emissions (0.89 Tg C yr$^{-1}$ and 0.2 Tg C yr$^{-1}$) are far lower than global emissions (440–660 Tg C yr$^{-1}$ and > 100Tg C yr$^{-1}$) (Yu and Li, 2021; Myriokefalitakis et

al., 2010; Zhang et al., 2025a; Byron et al., 2022; Guenther et al., 2006), the impact of local marine emissions still needs to be considered. Especially due to the input of terrestrial nutrients, coastal waters are the regions with the highest marine productivity in marginal seas. This is reflected by high Chl–a concentration on the sea surface near the coastline (Fig. S5). Hence, the biological emission of coastal waters is much stronger than that of sea regions far away from land. Besides, the biogenic SOA detected in this study primarily exist as CHONS compounds, indicating that nitrogen chemistry (involving NO$_x$

and NO$_3$) and sulfur chemistry (involving sulfate seeds) play important roles in SOA formation over marginal seas, which highlights that land–sea interaction can effectively change the chemical composition of aerosols over marginal seas.

Identified diesel-derived SOA molecules included $C_9H_{17}O_6S^-$, $C_8H_{15}O_6^-$, $C_8H_{17}O_4S^-$, and $C_8H_7O_3S^-$ (Wang et al., 2018; Wang et al., 2021; Hughes and Stone, 2019), primarily found in HULIS. The Bohai Rim region is one of the three major port clusters in China, with cargo throughput and container throughput accounting for 18.9% and 16.5% of the country throughputs,

respectively, and the impact of ship emissions on the atmosphere cannot be ignored. For example, previous studies have shown that ship emissions have a significant impact on coastal cities: the emissions of SO$_2$, NO$_x$, and PM$_{2.5}$ from ships account respectively for 10%, 8.6%, and 3% of the total anthropogenic emissions in Tianjin (Zhang et al., 2017; Chen et al., 2016). Hence, these diesel-derived SOA molecules in this study may originate from ship emissions over the sea or transportation emissions from land. Furthermore, the highly aromatic and unsaturated CHOS compounds only detected in the northern sea

region may reflect a unique local source, such as offshore drilling platforms. Numerous unsaturated oxygenated fatty acid compounds, generally with carbon numbers less than 22 and DBE ≤ 5, were present in HULIS from both sea regions (Fig. S6). Their water solubility is likely related to the presence of polar functional groups such as carboxyl and hydroxyl groups in their structures. Previous studies showed that significant amounts of polyunsaturated fatty acids (3–200 μg L$^{-1}$) exist on the sea surface, primarily produced by marine organisms, and containing multiple double bonds and long carbon chains (~22) (Novak

and Bertram, 2020; Gašparović et al., 2007; Colombo et al., 2017).

However, most aliphatic compounds detected in this study have lower carbon numbers and do not appear to be polyunsaturated fatty acids directly emitted from the ocean (Fig. S10). Instead, they seem to be secondary transformation products of polyunsaturated fatty acids. For example, polyunsaturated fatty acids can react with ozone at the air-sea interface to generate

lower carbon-number unsaturated fatty acids (Zhou et al., 2014). Alkanes and cycloalkanes in unburned fuel and lubricating oil could also be potential sources of these unsaturated fatty acids. In the atmosphere, these long-chain alkanes can be modified by oxidation and functionalization with the introduction of carbonyl and hydroxyl groups on hourly timescales (Tao et al., 2014; Yee et al., 2013), which can be potential sources of these unsaturated aliphatic compounds. Although it is currently difficult to distinguish whether these unsaturated fatty compounds come from ship emissions or terrestrial transport, the hydrophobicity conferred by long carbon chains should be considered because of their possible impact on aerosol hygroscopicity and cloud condensation nuclei activity.

Characteristic molecules in WISOC from aerosols in both sea regions were primarily anthropogenic substances with high saturation. These included anthropogenic antioxidants, alkylbenzene sulfonates, and relatively saturated aliphatic substances. Anthropogenic antioxidants were mainly $C_{23}H_{31}O_2^-$ and its derivatives ($C_{23}H_{31}O_3^-$, $C_{23}H_{29}O_3^-$). Based on its MS/MS spectrum, $C_{23}H_{31}O_2^-$ was identified as 2,2'-methylenebis(6-tert-butyl-4-methylphenol) (Fig. S11A and B). This compound, used as an antioxidant, is widely applied in the rubber industry, synthetic materials, and petroleum products to prevent oxidation. It also possesses light-absorbing properties, and its impact on marine atmospheric radiative forcing and marine ecosystems needs future investigation. Alkylbenzene sulfonates are anthropogenic surfactants widely used in synthetic detergents and personal care products (Li et al., 2024). For example, alkylbenzene sulfonates detected in this study are branched and difficult to degrade, belonging to persistent environmental pollutants (Fig. S12). Previous studies reported their widespread presence in aerosols, rainwater, seawater, and rivers (Chen et al., 2024; Cochran et al., 2016; Cai et al., 2024; Ma et al., 2024). The BS and YS are significantly affected by the transport of nearby rivers (Liu et al., 2020), with the annual inflow of the Yellow River alone reaching $390.88 \times 10^8$ m$^3$. Therefore, anthropogenic pollutants can enter BS and YS through rivers in coastal terrestrial regions and reenter the atmosphere via sea spray. The relatively saturated aliphatic compounds were dominated by molecules containing O/N $\geq$ 3, which suggests that they are most likely secondary products from reactions of unsaturated hydrocarbons with NO$_x$.

### 3.4 Similarity of aerosol light absorption

The light–absorbing components in marginal sea aerosols primarily originate from land, and their light–absorbing capacity diminishes during long–range transport due to photobleaching and aging. To further investigate the differential influence of inland versus coastal terrestrial emissions on BS and YS summer aerosols, we analyzed the optical properties of characteristic light-absorbing components. Our results show that there is no apparent difference in the light–absorbing parameters (Abs$_{365}$, MAE$_{365}$, AAE) of typical light absorption components between the two sea regions (Table S5). The mass absorption efficiency at 365 nm (MAE$_{365}$) is a key parameter for assessing aerosol light absorption capacity. Typically, freshly emitted light–absorbing components from source regions exhibit higher MAE$_{365}$. The similarity in MAE$_{365}$ for the same organic components between the two regions is a hint that they may originate from similar source regions or have undergone similar atmospheric chemical processes. Fluorescence component analysis also revealed that the types and proportions of identified fluorescent

components in both water-soluble and water-insoluble compounds were similar between the two regions (Fig. 3A and Fig. S13).

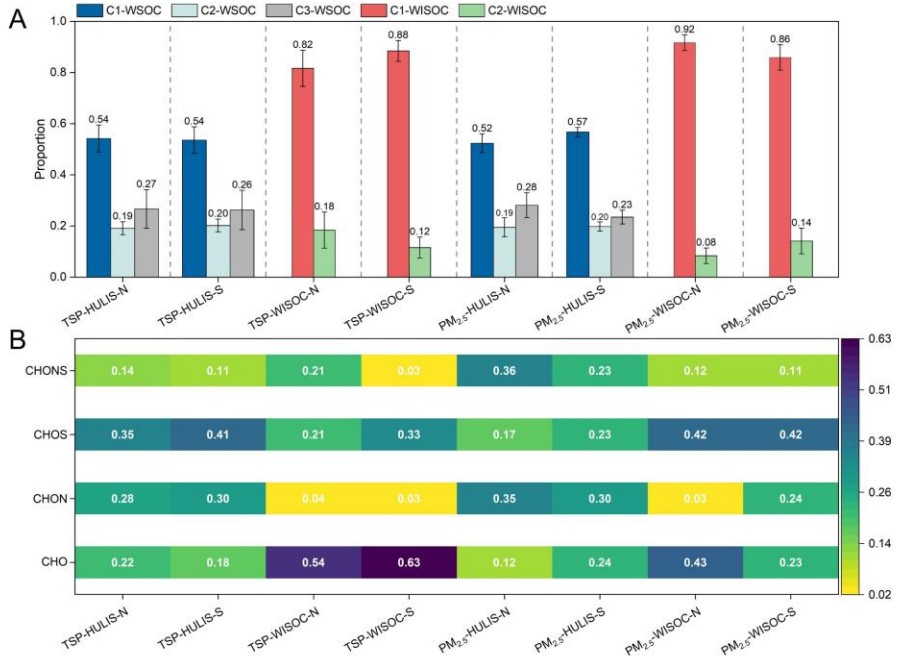

**Figure 3. (A) Proportion of fluorescent components in WSOC and WISOC. C1–WSOC and C2–WSOC are two humic–like (HULIS) components in WSOC, C3–WSOC is the protein–like (PRLIS) component in WSOC. C1–WISOC and C2–WISOC are HULIS and PRLIS components in WISOC, respectively. (B) Proportion of four types of potential light–absorbing organic compounds in different types of particles.**

The major light–absorbing peaks occurred within the first two minutes of gradient elution (Fig. S14). Previous studies suggested that molecules with DBE/C $\geq$ 0.5 are potentially light-absorbing (Huang et al., 2020). Analysis from the perspective of potential light–absorbing molecular composition showed a similarity in the proportions of light–absorbing components between the two regions. The intensity-weighted parameters for molecules meeting this criterion within the light–absorbing peaks are shown in Table S6. Compared to the overall molecular characteristics of the samples, these potential light-absorbing molecules exhibited higher DBEw, AIw, and carbon oxidation state, indicating that they are more unsaturated, aromatic, and oxidized. The proportional variation of four classes of compounds within the potential light-absorbing molecules is shown in Fig. 3B. Overall, the proportions of these four classes showed little difference (around 10%) between aerosols from the northern and southern sea regions, although the proportion of certain classes of compounds in WISOC differed significantly. Previous studies have shown that the photolysis half-life of light–absorbing components in aerosols is 9–15 h, and the average atmospheric lifetime of all light–absorbing components is about 20 h (Forrister et al., 2015; O'brien et al., 2025). Comparing the $MAE_{365}$ of WSOC in summer aerosols of BS and YS with coastal cities (Table S7), we found that the $MAE_{365}$ of BS and YS aerosols is about 50% of that in coastal cities. Moreover, the transport time of air masses from the land to the ocean is

mainly within 30 h (Fig. S15), which is consistent with the half-life of organic component photolysis. This indicates that coastal terrestrial regions are the most likely sources of summer BS and YS aerosols. Furthermore, the air mass mainly transports boundary layer material within 190–260 km from the coastline (Fig. 2B and C). Hence, we speculate that components in aerosols over BS and YS are more likely to originate from coastal terrestrial regions rather than inland.

**3.5 Source apportionment based on $\delta^{13}C_{TC}$ and positive matrix factorization**

In order to ensure that the dataset is large enough to generate more reliable results for PMF, we integrated all TSP and $PM_{2.5}$ samples into one dataset, and results are shown in Fig. 4A. Five factors were identified as the optimal solution. The robustness of PMF results and potential collinearity between factors have been discussed in detail in the Text S3 of the Supplement. Briefly, the five–factor solution has a low Q(True)/Q(Robust) ratio. Two error estimation methods (DISP and BS) jointly reveal that there is no factor swap in five factors, and the matching rates of five factors are close to 100%. The scaled residuals of each species are generally within + 3 and - 3, and G–plot reveals a weak collinearity between factors. Therefore, the five–factor solution is relatively robust.

The species with a high proportion in the profile of factor 1 is $Ca^{2+}$ or $nss–Ca^{2+}$, which is commonly believed to originate from the crust or soil (Stanimirova et al., 2023). This factor is identified as a crustal source. The characteristic ion components in factor 2 are $Na^+$, $Mg^{2+}$ and $Cl^-$, with $Na^+$ and $Cl^-$ exhibiting the highest proportion (Zong et al., 2016). This factor may then be associated with sea salt and is identified as a marine primary source. Factor 3 has a high proportion of secondary inorganic ions ($NH_4^+$ and $NO_3^-$) derived from heterogeneous or homogeneous reaction of $NH_3$ and $NO_2$ (Pathak et al., 2009), and it is considered as a secondary inorganic source (Wei et al., 2024). Factor 4 has high proportions of EC, organic species, $nss–K^+$ and $nss–SO_4^{2-}$. EC and $nss–K^+$ jointly indicate that this factor may have connection with combustion source (biomass burning), while $nss–SO_4^{2-}$ is associated with secondary transformation of $SO_2$ (Dai et al., 2023; Xue et al., 2019). Hence, factor 4 is identified as a mixture source (combustion source and $SO_4^{2-}$). Factor 5, with a high proportion of SOC and low proportions of EC and inorganic species, is considered as a secondary organic source.

The above proportions are consistent with results of previous studies conducted at BS and YS that have also shown that biomass combustion, secondary sources, and sea salt are common sources of aerosol components (Zhao et al., 2023; Geng et al., 2020; Zhang et al., 2025b). Among them, combustion and secondary sources (secondary organic and inorganic aerosol sources) constituted the largest proportion (> 60%). This highlights the importance of combustion source and atmospheric secondary transformation in the atmosphere of marginal seas. Additionally, the proportion of marine primary sources was relatively low in the northern sea region, while dust and secondary organic aerosol (SOA) sources were higher (Fig. S16), indicating that low Rtbl can still transport significant amounts of terrestrial components.

Due to the lack of typical marine organic tracers, PMF cannot separate the contribution of marine organic aerosol sources alone and can only identify the contribution of marine primary sources. But the $\delta^{13}C_{TC}$ of organic components from terrestrial and marine sources are different. Hence, $\delta^{13}C_{TC}$ can be used as additional evidence to assess the impact of marine organic sources when there is lack of organic tracers in PMF model. The average $\delta^{13}C_{TC}$ value of carbonaceous components in aerosols

ranged from –25.9‰ to –24.6‰, with an average of -25.2‰ (–25‰ for TSP, –25.3‰ for PM$_{2.5}$). These values are closer to the characteristic carbon stable isotope signatures of coal combustion and biomass burning aerosols (–24‰ to –28.4‰) and lower than those of typical marine sources (–18‰ to –23‰) (Bikkina et al., 2022; Crocker et al., 2020).

Combining above results with a Bayesian mixing model, we found that carbonaceous components in BS and YS summer aerosols were predominantly influenced by biomass burning sources, primarily from C3 plant burning, contributing approximately 60–80% to the carbonaceous fraction. Biomass burning remained dominant even when accounting for isotopic fractionation effects (Fig. 4B–E). The high contribution of biomass burning may have connection with dense open fire points in coastal terrestrial regions. In general, marine emission only contribute less than 10% of carbonaceous species. However, due to the different principles of the two models, it is difficult to match the δ $^{13}$C$_{TC}$ results with the PMF results. Nethertheless, it can be speculated that in the PMF model, the contribution of marine emissions to organic aerosols may be merged into other sources containing OC or SOC, such as secondary organic aerosol source. In the future, it is necessary to include typical marine organic tracers (such as methanesulfonic acid) to the PMF model for more accurate source apportionment.

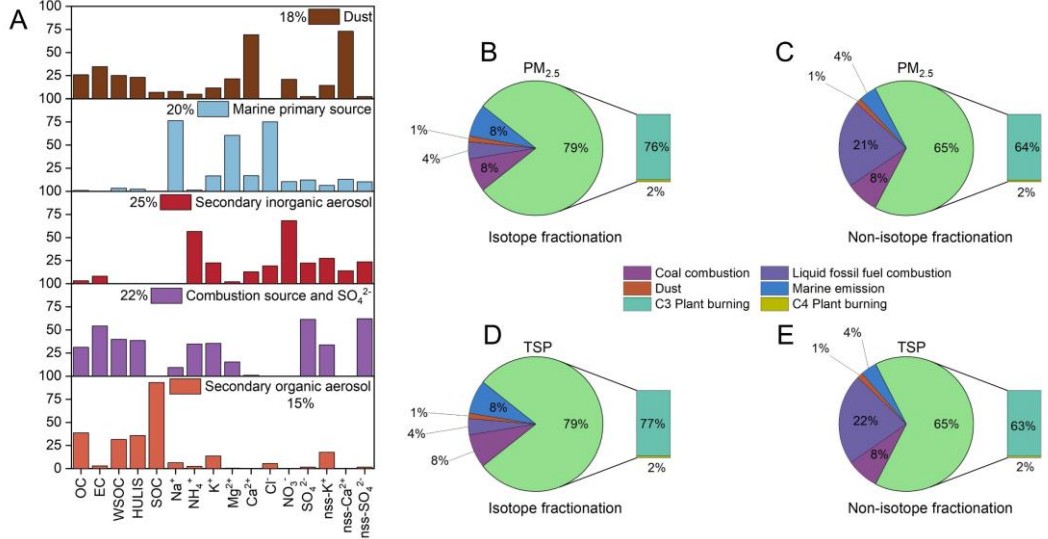

Figure 4. (A) Source apportionment based on PMF. (B) to (E) Source apportionment based on δ$^{13}$C$_{TC}$.

## 4. Conclusion

Our results indicate that the proportion of inland air masses reaching BS and YS in summer decreases, while the proportion of coastal and marine air masses increases. The air masses reaching over BS and northern YS originate inland, but have a high initial height (>1000 m) and a low Rtbl (0.40 ± 0.26), mainly transport terrestrial components within the boundary layer of coastal provinces. The air masses reaching the southern YS mainly transport within coastal boundary layer and marine boundary layer. Overall, the two sea regions are still dominated by oceanic boundary layer air masses (> 80%). In terms of chemical composition, the ratios of OC/EC, WSOC/OC, and HULIS/WSOC all show obvious terrestrial characteristics, and

non–sea salt components are the main components of water-soluble ions (> 80%), much higher than the proportion of sea salt. Coastal dense fire points, source apportionment based on $\delta^{13}C_{TC}$ and PMF jointly revealed that biomass burning in coastal provinces contributes significantly to carbonaceous components (60%–80%), and the contribution of marine primary source only accounts for 20%. Among all typical molecules, anthropogenic molecules dominate: oxygen-containing aromatic compounds, nitroaromatic compounds or unsaturated hydrocarbons containing nitro groups, fossil fuel SOA, anthropogenic surfactants (alkylbenzene sulfonates), anthropogenic antioxidants. Biogenic SOA are relatively few, being basically monoterpenes and isoprene SOA. But some unsaturated fatty compounds may have both anthropogenic sources (traffic emissions) and marine sources (unsaturated and polyunsaturated fatty acids). Although the sources of air masses in the northern and southern sea areas are different, there are similarities in the aerosol absorption components. The time it takes for air masses to be transported from land to the ocean in this study (< 30 h) and the absorbance of WSOC in BS and YS aerosols is about 50% of that in coastal cities. This is consistent with the half-life of the absorbing components reported in the literature (< 20 h), suggesting that BS and YS aerosols are more likely to be emitted from coastal provinces. The height variation of air masses further reveals the dominant role of terrestrial emissions primarily from coastal provinces (the Bohai Rim region within 190–260 km from the coastline), rather than long–range transport from inland.

From a global perspective, coastal terrestrial regions (usually within 100 km of the coastline) accommodate nearly one-third of the global population with a relatively small land area (18%), contribute nearly 82% of the world's gross domestic product (GDP) and contain 67% of mega cities (Jin et al., 2023; Reimann et al., 2023; Von Glasow et al., 2012). Anthropogenic emissions from these regions dominate on a global and time scale. Therefore, even in winter and spring seasons, anthropogenic emissions from coastal terrestrial regions should be dominant for the atmospheric environment of marginal seas. However, because most of the air masses reaching BS and YS during the cold season originate inland, the term "long–range transport" is commonly used to describe the transport of inland pollutants to marginal seas. This may, to some extent, mask the important role of emissions in coastal regions. Our results emphasize the importance of coastal terrestrial emissions in controlling marginal sea environmental pollution by quantifying the range of coastal regions that affect the atmospheric environment of BS and YS. Controlling coastal terrestrial emissions is an important measure to alleviate marginal sea environmental pollution. The results of this study are also applicable to other marginal seas around the world, providing a reference for exploring the potential sources, transport, and transformation of marine atmospheric pollutants. At the same time, it also suggests that when using regional climate chemistry coupling models to simulate land–sea interactions, special attention should be paid to updating coastal terrestrial emissions.

**Data availability**

The data from this research can be obtained upon request by contacting the corresponding author.

## Author contributions

Conceptualization: LD and JL; Funding acquisition: LD; Investigation: KH; Supervision: LD and JL; Writing – original draft preparation: KH. Writing – review & editing: NTT, KL, HH, JL and LD.

## Competing interests

The authors declare that they have no conflict of interest.

## Financial Support

This work was supported by National Natural Science Foundation of China (22376121, 22361162668, 42006044 and 42476033) and Intramural Joint Program Fund of State Key Laboratory of Microbial Technology (SKLMTIJP-2025-02). Data and samples were collected onboard of R/V "Lanhai101" implementing the open research cruise NORC2023–01 supported by NSFC Ship time Sharing project (project number: 42249901).

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
