# Peer review of "Coastal terrestrial emissions modify the composition and optical properties of aerosols in marginal seas"

_EGUsphere, 2025_

## Author Comment (AC1)

**Reply to Anonymous Referee #4**

We thank Anonymous Referee #4 for the valuable comments on our manuscript. Here we provide point-to-point responses to the Referees' comments. For clarity, the Referees' comments are marked in black, authors' responses are marked in **blue**, and changes in the manuscript are marked in **red**.

This investigated the chemical compositions and properties of marine aerosols over the Bohai Sea and Yellow Sea. This work is generally well-written and within the scope of ACP. I support the publication once the following comments are addressed.

1. For the PMF source apportionment analysis, I'm confused by the statements in lines 346-350. Have the authors investigated the uncertainty of your PMF results? This is needed before discussion based on the results. Sometimes, terrestrial sources and anthropogenic pollutants occur at the same time. Have the authors check the collinearity between sources (e.g., secondary inorganic aerosols, combustion, secondary organic aerosols, dust)?

**Author reply:**

Concerning the statement at lines 346–350, our original idea was to express that combining TSP and $PM_{2.5}$ samples for PMF source apportionment yields better results than conducting source apportionment separately for TSP and $PM_{2.5}$, because the larger the dataset, the more reliable the PMF parsing results are. We have rewritten the sentence in the manuscript as follows (Page 14, Line 362–363):

In order to ensure that the dataset is large enough to generate more reliable results for PMF, we integrated all TSP and $PM_{2.5}$ samples into one dataset, and results are shown in Fig. 4A.

Regarding the uncertainty in the PMF results, we have added additional error estimation analysis, residual analysis, and fitting coefficients of each species in the model. Table R1 shows the two error estimation methods: Displacement (DISP) and Bootstrap (BS).

The fact that no factors swaps are observed in DISP result this indicates no significant rotational ambiguity and a relatively robust solution. BS results show that although not all of the base factors were mapped to the boot factors, matching rates of all factors are close to 100%, with only one factor having a low matching rate that still exceeds 80%. This indicates that the five–factor solution is relatively stable. The unmapped factors may be due to the combination of the high variability in the data and PMF not fitting all of the data.

We also verified the error evaluation results of the four–factor and six–factor solutions. The matching rate of each of these solutions (Table R2 and Table R3) is lower than that of the five–factor solution, and their Q(True)/Q(Robust) ratios are higher than that of the five–factor solution (Figure R1). Low Q(True)/Q(Robust) ratio is usually indicative of the reasonableness of the model results (Song et al., 2018b). Low Q(True)/Q(Robust) ratio and high matching rate indicate that five factors are more suitable than four factors. In addition, the fitting coefficients of each species in the five–factor solution are high enough (Table R4). The scaled residuals of all species in the five–factor solution are mainly within + 3 and – 3 (Figure R2), indicating that each species fits well in the model. Hence, we finally report the results of the five–factor solution.

Table R1. Results of two error estimation methods (Displacement: DISP, Bootstrap: BS) for five–factor solution.

| DISP Diagnostics | | | | | |
|---|---|---|---|---|---|
| Error Code: | 0 | | | | |
| Largest Decrease in Q: | -0.047 | | | | |
| %dQ: | -0.0071 | | | | |
| Swaps by Factor: | 0 | 0 | 0 | 0 | 0 |
| BS Mapping | | | | | |
| | Base Factor 1 | Base Factor 2 | Base Factor 3 | Base Factor 4 | Base Factor 5 | Unmapped |
| Boot Factor 1 | 96 | 0 | 2 | 1 | 1 | 0 |
| Boot Factor 2 | 0 | 98 | 0 | 0 | 0 | 2 |
| Boot Factor 3 | 1 | 0 | 99 | 0 | 0 | 0 |
| Boot Factor 4 | 1 | 3 | 3 | 89 | 3 | 1 |
| Boot Factor 5 | 0 | 0 | 0 | 0 | 100 | 0 |

Table R2. Results of two error estimation methods (Displacement: DISP, Bootstrap: BS) for four–factor solution.

| DISP Diagnostics | | | | |
|---|---|---|---|---|
| Error Code: | 0 | | | |
| Largest Decrease in Q: | -0.056 | | | |
| %dQ: | -0.0044 | | | |
| Swaps by Factor: | 0 | 0 | 0 | 0 |
| BS Mapping | | | | |
| | Base Factor 1 | Base Factor 2 | Base Factor 3 | Base Factor 4 | Unmapped |
| Boot Factor 1 | 95 | 0 | 0 | 4 | 1 |
| Boot Factor 2 | 2 | 90 | 2 | 2 | 4 |
| Boot Factor 3 | 4 | 3 | 89 | 3 | 1 |
| Boot Factor 4 | 4 | 1 | 0 | 94 | 1 |

Table R3. Results of two error estimation methods (Displacement: DISP, Bootstrap: BS) for six–factor solution.

| DISP Diagnostics | | | | | | |
|---|---|---|---|---|---|---|
| Error Code: | 0 | | | | | |
| Largest Decrease in Q: | -0.011 | | | | | |
| %dQ: | -0.030 | | | | | |
| Swaps by Factor: | 0 | 0 | 0 | 0 | 0 | 0 |
| BS Mapping | | | | | | |
| | Base Factor 1 | Base Factor 2 | Base Factor 3 | Base Factor 4 | Base Factor 5 | Base Factor 6 | Unmapped |
| Boot Factor 1 | 97 | 0 | 2 | 1 | 0 | 0 | 0 |
| Boot Factor 2 | 4 | 71 | 5 | 5 | 3 | 9 | 3 |
| Boot Factor 3 | 5 | 2 | 84 | 3 | 2 | 4 | 0 |
| Boot Factor 4 | 4 | 9 | 2 | 78 | 5 | 2 | 0 |
| Boot Factor 5 | 0 | 0 | 0 | 0 | 100 | 0 | 0 |
| Boot Factor 6 | 0 | 1 | 0 | 0 | 0 | 99 | 0 |

Table R4. Fitting results between observed concentration and predicted concentration of each species in the five–factor solutions.

| | Four–factor solution | Five–factor solution | Six–factor solution |
|---|---|---|---|
| Species | $r^2$ | $r^2$ | $r^2$ |
| Particle | 0.70 | 0.70 | 0.70 |

| | | | |
|---|---|---|---|
| OC | 0.86 | 0.94 | 0.97 |
| EC | 0.55 | 0.72 | 0.76 |
| WSOC | 0.96 | 0.97 | 0.98 |
| HULIS | 0.93 | 0.93 | 0.97 |
| SOC | 0.75 | 0.98 | 0.97 |
| $Na^+$ | 0.97 | 0.98 | 0.99 |
| $NH_4^+$ | 0.98 | 0.98 | 0.98 |
| $K^+$ | 0.83 | 0.82 | 0.95 |
| $Mg^{2+}$ | 0.97 | 0.98 | 0.99 |
| $Ca^{2+}$ | 0.97 | 0.99 | 1.00 |
| $Cl^-$ | 0.95 | 0.97 | 0.98 |
| $NO_3^-$ | 0.92 | 0.93 | 0.89 |
| $SO_4^{2-}$ | 0.82 | 0.91 | 0.99 |
| $nss–K^+$ | 0.80 | 0.79 | 0.93 |
| $nss–Ca^{2+}$ | 0.96 | 0.99 | 1.00 |
| $nss–SO_4^{2-}$ | 0.82 | 0.91 | 0.98 |

[Figure]

Figure R1. Variation of Q (True)/Q (Robust) with the increase of factor number. The red circles represent the optimal factor number.

[Figure]

Figure R2. Residual distribution of each species in the five–factor solution.

Given the above error estimation results, it can be proven that the five–factor solution is more robust. Following this, we will mainly discuss the collinearity of the five–factor solution. Our results indicate that the collinearity among the five factors is weak, which

can be demonstrated by the following aspects:

Firstly, DISP results show that there is no factor swap between different factors, and BS results show high factor mapping rates (Table R1). These all suggest that the collinearity between factors is weak.

Secondly, we examined the G–plot between five factors (Figure R3). In theory, when there is no collinearity between factors, sample points should be close to or near the coordinate axis, that is, $x = 0$ or $y = 0$. If sample points are far from the coordinate axis or show an approximately correlated straight line, it indicates high collinearity between factors and high rotational uncertainty. Figure R3 shows that for most factors, G–plot shows that most samples points are located near the coordinate axis, indicating that they represent meaningful source contributions rather than redundant or collinear factors. In addition, there is also no significant linear correlation between factors. For mixed sources (combustion source + $SO_4^{2-}$), the G–plot shows that the partial sample points are far away from the coordinate axis. For example, there is a certain linear trend in the G–plot between the mixed source and dust, and partial sample points between the mixed source and the secondary inorganic source are also far away from the coordinate axis. As pointed out by the Referee, terrestrial sources and anthropogenic pollutants occur at the same time. Therefore, the weak collinearity between these three factors may be due to the existence of shared emission zones or atmospheric chemical processes among these sources, such as dust and biomass combustion both coming from land, and $SO_4^{2-}$, $NH_4^-$ and $NO_3^-$ all coming from atmospheric transformation. In summary, our results (DISP, BS, and G-plot) explain that the collinearity of the five–factor solution is relatively weak and acceptable.

[Figure]

Figure R3. G–plot of five–factor solution.

Based on the Referee's suggestion, before formally discussing the PMF results, we briefly added the following content in the revised manuscript and Supplement to illustrate their robustness.

Main manuscript, Page 14, Line 363–338:

Five factors were identified as the optimal solution. The robustness of PMF results and potential collinearity between factors have been discussed in detail in the Text S3 of the Supplement. Briefly, the five–factor solution has a low Q(True)/Q(Robust) ratio. Two error estimation methods (DISP and BS) jointly reveal that there is no factor swap in five factors, and matching rates of five factors are close to 100%. The scaled residuals of each species are generally within + 3 and - 3, and G–plot reveals a weak collinearity between factors. Therefore, the five–factor solution is relatively robust.

Text S3 of the Supplement, Page 6, Line 112–135:

The collinearity problem between factors is not only affected by the number of factors and the lack of typical tracers, but also by co–emission, co–transport and secondary transformation between sources. The results of DISP and BS jointly reveal that the factor swap does not occur and the factor matching rate is high. This indicates a low

possibility of collinearity caused by the number of factors or tracers. To further evaluate the collinearity caused by co–emission, co–transport and secondary transformation, we compared the G-plots of five factors (Figure S19). In theory, when there is no collinearity between factors, sample points should be close to or near the coordinate axis, that is, $x = 0$ or $y = 0$. If sample points are far from the coordinate axis or show an approximately correlated straight line, it indicates high collinearity between factors and high rotational uncertainty. Figure S19 shows that for most factors, G–plot shows that most samples exhibit non–zero or near zero contributions, with many points located near the coordinate axis, indicating that they represent meaningful source contributions rather than redundant or collinear factors. In addition, there is no significant linear correlation between factors. For mixed sources (combustion source + $SO_4^{2-}$), the G–plot shows that the partial sample points are far away from the coordinate axis. For example, there is a certain linear trend in the G–plot between the mixed source and dust, and partial sample points between the mixed source and the secondary inorganic source are also far away from the coordinate axis. The weak collinearity between these three factors may be due to the existence of shared emission zones or atmospheric chemical processes among these sources, such as dust and biomass combustion both coming from land, and $SO_4^{2-}$ $NH_4^-$ and $NO_3^-$ all coming from atmospheric transformation. In summary, our evaluation results (DISP, BS, and G-plot) explain that the collinearity of the five–factor solution is relatively weak and acceptable.

[Figure]

Figure S19. G–plot of five–factor solution.

1. It seems the authors focused on the results of $\delta^{13}C_{TC}$ to analyze the sources of marine organic aerosols. How about the source contributions of marine organic aerosols apportioned by the PMF model? I may suggest to add related discussion in section 3.5.

**Author reply:**

Due to the lack of specific tracers for marine organic aerosols (MOA) in our dataset, PMF was not able to resolve an independent MOA factor. The contribution of marine organic aerosols may be merged into other sources containing OC or SOC, such as secondary organic aerosol source. $\delta^{13}C_{TC}$ provides complementary source information that is less dependent on molecular–level tracers and is particularly useful for distinguishing organic compounds from marine source and combustion–related source. Therefore, when the lack of organic tracers from marine source makes it impossible to analyze the contribution of marine organic aerosols separately through PMF, $\delta^{13}C_{TC}$ can be used as additional evidence to assess the impact of marine organic sources.

We have added a brief discussion in Section 3.5. At the same time, we have adjusted the structure of this section by first introducing the results of PMF, clarifying the limitations of PMF in analyzing the contribution of marine organic aerosol sources in

the absence of marine organic aerosol tracers. Continuing with the introduction of the source apportionment results of $\delta^{13}C_{TC}$, it is clarified that $\delta^{13}C_{TC}$ can serve as a supplement to PMF results, and the potential correlation between the results of $\delta^{13}C_{TC}$ and PMF is briefly explained. The following is the revised content of Section 3.5 (Page 14, Line 362–400) in the revised manuscript:

[revised manuscript text omitted]

2. This work highlights the importance of coastal emissions on the marine aerosols over the marginal seas. What do you mean "coastal emissions" here? It is unclear. Please specific in the title and throughout the manuscript. The "properties" in the title is unclear. Pleas be specific.

**Author reply:**

We have revised the title and the entire manuscript regarding the expression of coastal emissions and properties. The description of coastal emissions in the entire manuscript has been revised to coastal terrestrial emissions.

Revised title: Coastal terrestrial emissions modify the composition and optical properties of aerosols in marginal seas

Further corrections in the revised manuscript.

Page 2, Line 51: Marginal seas are adjacent to coastal terrestrial regions with important human activities and high anthropogenic emissions.

Page 8, Line 197: The orange region represents the coastal terrestrial regions of China.

Page 11, Line 279: The precursors of these biogenic SOA are most likely isoprene and monoterpenes released by terrestrial plants in coastal terrestrial regions and

phytoplankton in coastal waters.

Page 12, Line 324: Therefore, anthropogenic pollutants can enter BS and YS through rivers in coastal terrestrial regions and reenter the atmosphere via sea spray.

Page 14, Line 359: This indicates that coastal terrestrial regions are the most likely sources of summer BS and YS aerosols.

Page 14, Line 360: Hence, we speculate that components in aerosols over BS and YS are more likely to originate from coastal terrestrial regions rather than inland.

Page 16, Line 425: From a global perspective, coastal terrestrial regions (usually within 100 km of the coastline) accommodate nearly one-third of the global population with a relatively small land area (18%), contribute nearly 82% of the world's gross domestic product (GDP) and contain 67% of mega cities.

Page 16, Line 429: Therefore, even in winter and spring seasons, anthropogenic emissions from coastal terrestrial regions should be dominant for the atmospheric environment of marginal seas.

Page 16, Line 431–432: Our results emphasize the importance of coastal terrestrial emissions in controlling marginal sea environmental pollution by quantifying the range of coastal terrestrial regions that affect the atmospheric environment of BS and YS.

Page 16, Line 432–433: Controlling coastal terrestrial emissions is an important measure to alleviate marginal sea environmental pollution.

Page 16, Line 435–437: At the same time, it also suggests that when using regional climate chemistry coupling models to simulate land–sea interactions, special attention should be paid to updating coastal terrestrial emissions.

3. The abbreviations (e.g., Rmbl, Rtbl, Rmam, Rtam) in Fig 1b are not easy to follow. And similar abbreviations appear a lot in the main text. I may suggest to use more concise and clear abbreviations throughout the manuscript, and define them clearly when they first appear.

**Author reply:**

The meanings of these four letters are:

R stands for Retention ratio

mbl stands for marine boundary layer

tbl stands for terrestrial boundary layer

mam stands for marine air masses

tam stands for terrestrial air masses

Full names of these abbreviations are provided in Section 2.4 and in the caption of Figure 1 in the revised manuscript. The specific modifications are as follows:

Page 5, Line 131: The retention ratios of terrestrial air masses (Rtam) and marine air masses (Rmam) are calculated and modified from the method proposed by previous studies.

Page 6, Line 146–147: Rtbl indicates the retention ratio of terrestrial boundary layer air mass. Rmbl indicates the retention ratio of marine boundary layer air mass.

[Figure]

**Figure 1. (A) Trajectories of air masses arriving at the Yellow Sea and Bohai Sea during the sampling period. The simulated air mass transport time is 72 h. Green, yellow and red points represent TSP, PM$_{2.5}$ samples and fire points, respectively. Fire point information comes from https://firms.modaps.eosdis.nasa.gov/map. The yellow dashed line represents the boundary between the northern and southern sea regions. (B) Retention ratio of air masses over land and ocean, as well as the retention ratio of boundary layer air masses. Rmbl**

stands for Retention ratio of marine boundary layer air masses, Rtbl stands for Retention ratio of terrestrial boundary layer air masses, Rmam stands for Retention ratio of marine air masses and Rtam stands for Retention ratio of terrestrial air masses. (C) Differences in HULIS/WSOC ratio at different sampling points. (D) Proportion of carbonaceous species and water–soluble ions in particles. The pie charts represent the proportion of sea salt ions and non–sea salt ions in the total ions of each sea region. N and S indicate the northern and southern sea regions, respectively.

4. Lines 186-187: Please explain this statement in detail. Do you have any related references?

**Author reply:**

Our original idea was to express that air masses flowing over the ocean spend more than 90% of their time within the boundary layer, while air masses flowing over land spend less than 60% of their time in the boundary layer (Figure 1B). Hence, the retention ratio of terrestrial boundary layer air masses (Rtbl) reaching both sea regions is lower than that of marine boundary layer air masses (Rmbl). Considering that atmospheric pollutants mainly constrained within the boundary layer, high Rmbl theoretically indicates that aerosols are more affected by marine emissions. Previous studies have used these two parameters to evaluate the impact of air mass transport on the atmosphere of the Yellow Sea, East China Sea, and Gulf of Aqaba (Zhou et al., 2023; Zhou et al., 2021; Yan et al., 2024).

We have rewritten the related sentence and added more references as follows:

Page 8, Line 199–202:

During the sampling period, we observed that the retention ratio of marine boundary layer air masses (Rmbl) in the two sea regions (> 90%) was higher than that of terrestrial boundary layer air masses (Rtbl) (< 60%) (Fig. 1B). Considering that atmospheric pollutants mainly constrained within the boundary layer, high Rmbl theoretically indicates that aerosols are more affected by marine emissions (Zhou et al., 2023; Zhou et al., 2021; Yan et al., 2024).

[Figure]

**Figure 1. (A) Trajectories of air masses arriving at the Yellow Sea and Bohai Sea during the sampling period. The simulated air mass transport time is 72 hours. Green, yellow and red points represent TSP, PM$_{2.5}$ samples and fire points, respectively. Fire point information comes from https://firms.modaps.eosdis.nasa.gov/map. The yellow dashed line represents the boundary between the northern and southern sea regions. (B) Retention ratio of air masses over land and ocean, as well as the retention ratio of boundary layer air masses. Rmbl stands for Retention ratio of marine boundary layer air masses, Rtbl stands for Retention ratio of terrestrial boundary layer air masses, Rmam stands for Retention ratio of marine air masses and Rtam stands for Retention ratio of terrestrial air masses. (C) Differences in HULIS/WSOC ratio at different sampling points. (D) Proportion of carbonaceous species and water–soluble ions in particles. The pie charts represent the proportion of sea salt ions and non–sea salt ions in the total ions of each sea region. N and S indicate the northern and southern sea regions, respectively.**

5. Lines 310-311: What do you mean "no significant differences in...." here? Please explain and be specific.

**Author reply:**

The term "no significant differences" here was used to express the fact that the optical parameters (Abs$_{365}$ and MAE$_{365}$) of the light–absorbing components (WSOC, HULIS, WISOC) in aerosols do not differ significantly between the northern and southern sea

regions, just as shown in Table R5 below. To make it clear, we have revised the sentence as follows:

Page 12, Line 331–332

Our results show that there is no apparent difference in the light–absorbing parameters ($Abs_{365}$ and $MAE_{365}$) of typical light–absorbing components between the two sea regions (Table S5).

Table R5. Major light absorption parameters of TSP and $PM_{2.5}$ in the two sea regions.

[revised manuscript text omitted]

---

## Author Comment (AC2)

**Reply to Anonymous Referee #3**

We thank Anonymous Referee #3 for the valuable comments on our manuscript. Here we provide point-to-point responses to the Referees' comments. For clarity, the Referees' comments are marked in black, authors' responses are marked in **blue**, and changes in the manuscript are marked in **red**.

Hu et al. investigate the chemical composition, optical properties, and sources of aerosols (TSP and PM$_{2.5}$) over the Bohai Sea (BS) and Yellow Sea (YS) during the summer of 2023. By combining bulk chemical analysis (ions, carbonaceous components), stable carbon isotopes ($\delta^{13}C_{TC}$), and high-resolution mass spectrometry (Q-TOF MS) for molecular characterization, the authors provide an insight on how coastal terrestrial emissions impact the marginal sea atmospheres. It is an interesting study and the paper is well-written in general. I have some minor comments on the manuscript.

1.  Title: I think that the authors have to revise the title of the manuscript to be more specific as 'coastal emissions' is too unclear and undefinable.

**Author reply:**

According to our research results, the term "Coastal emission" in the manuscript is intended to highlight terrestrial emissions from coastal provinces. Therefore, we ultimately revised the title to:

Coastal terrestrial emissions modify the composition and optical properties of aerosols in marginal seas

2.  Line 23-24, Abstract: Suggest to revise the sentence. It is not clear how terrestrial emissions from coastal regions remain the major factor affecting marginal aerosols. Please be specific.

**Author reply:**

We have made revisions to the last sentence of the Abstract, emphasizing the terrestrial characteristics of aerosols and the sources of air masses (Page 1, Line 23–26).

Our results emphasize that during summer when the influence of marine air masses increases, the terrestrial characteristics of BS and YS aerosols remain evident, being related to air mass transport from coastal terrestrial regions.

3. Line 61: The term "weekly" seems inappropriate here. Did the authors mean "weakly"?

**Author reply:**

We thank the Referee for highlighting this error. It has been corrected to "weakly" in the revised manuscript. Page 2, Line 62–64:

In the BS and the northern YS heavily polluted by terrestrial sources, the formation pathway of atmospheric $NO_3^-$ is dominated by anthropogenic hydrocarbon, which is significantly different from that in the southern YS, weakly influenced by terrestrial activities.

4. Lines 73-75: Sampling on the first deck of a research vessel is highly susceptible to the influence of the ship's own exhaust plumes. If any measures were taken to exclude self-contamination (e.g., wind sector control or BC filtering), please describe them in detail.

**Author reply:**

Placing the sampler on the first deck is currently a common practice for collecting field aerosols. The purpose of doing so is to keep the sampler as far away from the sea surface as possible, to avoid direct impact of splashing seawater on samples, and also to avoid the influence of ship exhaust at the stern. As for the wind sector control or BC filtering mentioned by the Referee, some studies have indeed adopted this method (Huang et al., 2022; Kim et al., 2015). However, this requires the sampler to be equipped with a wind direction monitoring device to prevent it from collecting airflow from the stern of the ship, or online aerosol mass spectrometer on board that can identify the sampling periods with high BC content. Here, we have taken another different preventive measure, namely the current common practice: instruments only collect samples during sailing (Song et al., 2018a; Zhao et al., 2024). Because only during the sailing of the

ship can it be ensured that the airflow comes as much as possible from the bow of the ship rather than the stern.

Details on avoiding the impact of ship emissions have been updated in the revised manuscript, Page 3, Line 78–79:

In order to avoid the impact of ship exhaust emissions and ensure that the collected airflow comes from the bow of the ship, the sampler only starts sampling when the ship is sailing.

5.   Lines 126-127: In Equations (5) and (6), 72 hours is used as the time scale for weight decay. What is the basis for selecting 72 hours? Should different weighting factors be applied for substances with different atmospheric lifetimes?

**Author reply:**

The atmospheric lifetime of aerosols is the key factor determining whether they can affect the receptor sites. The selection of 72 hours as the transport time for air masses is mainly based on two basic reasons:

Firstly, previous studies showed that the atmospheric lifetime of aerosol chemical composition mainly varies on time scales of hours to a few days (Gao et al., 2022a). For example, early research suggested that the atmospheric lifetime of aerosols ranged from 4 to 60 days (Giorgi and Chameides, 1986), with aerosols confined in source regions having short atmospheric lifetimes and aerosols undergoing long–range transport having a longer atmospheric lifetime (Balkanski et al., 1993). Although there are differences in the evaluation results of different models for the atmospheric lifetime of organic and inorganic aerosols, there is relatively little difference in the time scale of atmospheric lifetime for the same type of aerosol/composition. The AeroCom Phase III model shows the global average lifetime of some inorganic components: nitrate at 2–7.8 days (mean of 5 days), ammonium at 1.9–9.8 days (mean of 4.3 days), sulfate at 0.86–7.6 days (mean of 4.5 days) (Bian et al., 2017; Park et al., 2004). For organic aerosols, the global average lifetime estimated based on the AeroCom Phase II model is 3.8–9.6 days (mean of 5.7 days) (Tsigaridis et al., 2014), while GEOS Chem simulated organic aerosol lifetime of about 4.9 and 5.8 days using two different

schemes (Pai et al., 2020). The more complex MAM7 reported lifetimes of 3.8 days for sulfate, 3.4 days for ammonium at, 5 days for primary organic matter (POM), 4.1 days for secondary organic aerosols, and 4.4 days for black carbon (Liu et al., 2012). Hence, using 72 hours (3 days) as the simulation time for air masses is generally appropriate. This time covers the atmospheric lifetimes of the above aerosol components or within their atmospheric lifetime.

Secondly, when conducting air mass trajectory analysis in marginal seas, current field studies usually set the simulation duration to 48 or 72 hours, though some studies even use longer simulation times (5 days or 10 days) (Mo et al., 2022; Budhavant et al., 2020; Li et al., 2023; Xu et al., 2025). Besides, previous studies using Eq. (7) and Eq. (8) in the main manuscript to calculate retention ratios and air masses also set the simulation duration to 72 hours (Zhou et al., 2023; Liu et al., 2022). Therefore, 72 hours was chosen for consistency with previous studies.

Finally, for the Referee's comment of applying different weighting factors, the original formula mainly aims to express that due to the diffusion and deposition of substances during transport, the longer the transport time, the weaker the impact on the recipient site. The calculation results are presented in the form of air mass retention ratio. If different time weighting factors are used for different substances, it will increase computational complexity. Besides, the atmospheric lifetimes of different substances are different. Calculating the air mass retention ratio of each substance separately may introduce significant uncertainties. Therefore, we uniformly used 72 hours as the time weighting factor to maintain consistency with literature data.

This has been updated in the revised manuscript, Page 6, Line 155–158:

Given that the average atmospheric lifetime of organic and inorganic substances in aerosols reported in literature is mainly around 3–6 days (Pai et al., 2020; Liu et al., 2012), and referring to the commonly used air mass simulation time (Cohen et al., 2015), we chose to simulate the trajectory of the air mass within 72 hours.

6. Lines 145-146: PMF models typically require a large sample size to ensure the stability of factorization. The manuscript notes that sampling occurred from July 15-23

and August 11-13, with TSP sampling durations of 12 h and PM5 of 24 h. This implies a relatively small total number of samples (roughly estimated at fewer than 30-40 valid samples). Resolving 4-5 factors with such a small sample size can lead to highly uncertain in the results. I suggest the author to include the results of residual analysis and other error analyses in the text.

**Author reply:**

Based on the Referee's comment, we have provided the error evaluation results and residual analysis of PMF in the Supplement. Table R1 shows the two error evaluation methods: Displacement (DISP) and Bootstrap (BS). The fact that no factors swaps are observed in DISP result indicates no significant rotational ambiguity and a relatively robust solution. BS results show that although not all of the base factors were mapped to the boot factors, matching rates of all factors are close to 100%, with only one factor having a low matching rate that still exceeds 80%. This indicates that the five–factor solution is relatively stable. The unmapped factors may be due to the combination of the high variability in the data and PMF not fitting all of the data.

We also verified the error evaluation results of the four–factor and six–factor solutions. The matching rate of each of these solutions (Table R2 and Table R3) is lower than that of the five–factor solution, and their Q(True)/Q(Robust) ratios are higher than that of the five–factor solution (Figure R1). Low Q(True)/Q(Robust) ratio is usually indicative of the reasonableness of the model results (Song et al., 2018b). Low Q(True)/Q(Robust) ratio and high matching rate indicate that five-factor solutions are more suitable than four-factors solutions. In addition, the fitting coefficients of each species in the five–factor solution are high enough (Table R4). The scaled residuals of all species in the five–factor solution are mainly within + 3 and – 3 (Figure R2), indicating that each species fits well in the model. Hence, we finally report the results of the five–factor solution.

Table R1. Results of two error estimation methods (Displacement: DISP, Bootstrap: BS) for a five–factor solution.

| DISP Diagnostics | |
| --- | --- |
| Error Code: | 0 |

| | | | | | |
|---|---|---|---|---|---|
| Largest Decrease in Q: | -0.047 | | | | |
| %dQ: | -0.0071 | | | | |
| Swaps by Factor: | 0 | 0 | 0 | 0 | 0 |

| | BS Mapping | | | | | |
|---|---|---|---|---|---|---|
| | Base Factor 1 | Base Factor 2 | Base Factor 3 | Base Factor 4 | Base Factor 5 | Unmapped |
| Boot Factor 1 | 96 | 0 | 2 | 1 | 1 | 0 |
| Boot Factor 2 | 0 | 98 | 0 | 0 | 0 | 2 |
| Boot Factor 3 | 1 | 0 | 99 | 0 | 0 | 0 |
| Boot Factor 4 | 1 | 3 | 3 | 89 | 3 | 1 |
| Boot Factor 5 | 0 | 0 | 0 | 0 | 100 | 0 |

Table R2. Results of two error estimation methods (Displacement: DISP, Bootstrap: BS) for a four–factor solution.

| DISP Diagnostics | | | | |
|---|---|---|---|---|
| Error Code: | 0 | | | |
| Largest Decrease in Q: | -0.056 | | | |
| %dQ: | -0.0044 | | | |
| Swaps by Factor: | 0 | 0 | 0 | 0 |

| | BS Mapping | | | | |
|---|---|---|---|---|---|
| | Base Factor 1 | Base Factor 2 | Base Factor 3 | Base Factor 4 | Unmapped |
| Boot Factor 1 | 95 | 0 | 0 | 4 | 1 |
| Boot Factor 2 | 2 | 90 | 2 | 2 | 4 |
| Boot Factor 3 | 4 | 3 | 89 | 3 | 1 |
| Boot Factor 4 | 4 | 1 | 0 | 94 | 1 |

Table R3. Results of two error estimation methods (Displacement: DISP, Bootstrap: BS) for a six–factor solution.

| DISP Diagnostics | | | | | | |
|---|---|---|---|---|---|---|
| Error Code: | 0 | | | | | |
| Largest Decrease in Q: | -0.011 | | | | | |
| %dQ: | -0.030 | | | | | |
| Swaps by Factor: | 0 | 0 | 0 | 0 | 0 | 0 |

| | BS Mapping | | | | | | |
|---|---|---|---|---|---|---|---|
| | Base Factor 1 | Base Factor 2 | Base Factor 3 | Base Factor 4 | Base Factor 5 | Base Factor 6 | Unmapped |
| Boot Factor 1 | 97 | 0 | 2 | 1 | 0 | 0 | 0 |
| Boot Factor 2 | 4 | 71 | 5 | 5 | 3 | 9 | 3 |
| Boot Factor 3 | 5 | 2 | 84 | 3 | 2 | 4 | 0 |

| | | | | | | | |
|---|---|---|---|---|---|---|---|
| Boot Factor 4 | 4 | 9 | 2 | 78 | 5 | 2 | 0 |
| Boot Factor 5 | 0 | 0 | 0 | 0 | 100 | 0 | 0 |
| Boot Factor 6 | 0 | 1 | 0 | 0 | 0 | 99 | 0 |

Table R4. Fitting results between observed concentration and predicted concentration of each species in the four to six–factor solutions.

| Species | Four–factor solution $r^2$ | Five–factor solution $r^2$ | Six–factor solution $r^2$ |
|---|---|---|---|
| Particle | 0.70 | 0.70 | 0.70 |
| OC | 0.86 | 0.94 | 0.97 |
| EC | 0.55 | 0.72 | 0.76 |
| WSOC | 0.96 | 0.97 | 0.98 |
| HULIS | 0.93 | 0.93 | 0.97 |
| SOC | 0.75 | 0.98 | 0.97 |
| $Na^+$ | 0.97 | 0.98 | 0.99 |
| $NH_4^+$ | 0.98 | 0.98 | 0.98 |
| $K^+$ | 0.83 | 0.82 | 0.95 |
| $Mg^{2+}$ | 0.97 | 0.98 | 0.99 |
| $Ca^{2+}$ | 0.97 | 0.99 | 1.00 |
| $Cl^-$ | 0.95 | 0.97 | 0.98 |
| $NO_3^-$ | 0.92 | 0.93 | 0.89 |
| $SO_4^{2-}$ | 0.82 | 0.91 | 0.99 |
| nss–$K^+$ | 0.80 | 0.79 | 0.93 |
| nss–$Ca^{2+}$ | 0.96 | 0.99 | 1.00 |
| nss–$SO_4^{2-}$ | 0.82 | 0.91 | 0.98 |

[Figure]

Figure R1. Variation of Q (True)/Q (Robust) with the increase of factor number. The red circles represent the optimal factor number.

[Figure]

Figure R2. Residual distribution of each species in the five–factor solution.

To demonstrate the reliability of the PMF results, we have added an evaluation description of the model results (see Text S3), as well as corresponding residual and

error estimation result (Figure S18, Table S10 and Table S11) in the Supplement.

Supplement, Page 6, Line 97–111:

As shown in Figure S17, the Q(True)/Q(Robust) ratio significantly decreases with the number of factors increasing to five, but it weakly decreases when the number of factors exceeds five. Besides, both error evaluation methods reveal that the five–factor solution is stable. No factors swaps were observed in the DISP result, which indicates no significant rotational ambiguity and that the solution is relatively robust (Table S10). BS results show that although not all of the base factors were mapped to the boot factors, matching rates of four factors are close to 100%, with only one factor having a low matching rate that still exceeds 80% (Table S10). This indicates that the five–factor solution is relatively stable. The unmapped factors may be due to the combination of the high variability in the data and PMF not fitting all of the data. In addition, the fitting coefficients ($r^2$) of most species in the five–factor solution are higher than 0.9 (Table S11). The scaled residuals of all species are mainly within + 3 and – 3 (Figure S18), indicating that each species fits well in the model. Hence, we finally report the results of five–factor solution.

7. Section 3.5: This manuscript will benefit more by further extending the discussion in this section. The current discussion is too simple and not detail enough. For example, what kind of combustion source and atmospheric secondary transformation in the atmosphere is plausible for influencing the marginal seas. Or further showing how important are these sources in this region. Are there any previous study supports the current PMF results and so on?

**Author reply:**

In this Section, we mainly analyzed in detail how to determine the source of aerosols based on the factor profile of PMF. Coming to the Referee's comment, we have identified that the combustion source may be from biomass, which is consistent with the analysis results based on $\delta^{13}C_{TC}$. As for the atmospheric secondary transformation, although we can determine which secondary species account for a larger proportion based on the factor profile, assessing the pathways through which these secondary

species are generated is beyond the scope of this study. Nevertheless, we briefly explored the possible formation pathways of secondary species reported in the literature. This has been updated in the revised manuscript, Page 14, Line 368–380:

The species with a high proportion in the profile of factor 1 is $Ca^{2+}$ or $nss–Ca^{2+}$, which is commonly believed to originate from the crust or soil (Stanimirova et al., 2023). Therefore, this factor is identified as a dust source. The characteristic ion components in factor 2 are $Na^+$, $Mg^{2+}$ and $Cl^-$, with $Na^+$ and $Cl^-$ exhibiting the highest proportion (Zong et al., 2016). Therefore, this factor may be associated with sea salt and is identified as a marine primary source. Factor 3 has a high proportion of secondary inorganic ions ($NH_4^+$ and $NO_3^-$) derived from heterogeneous or homogeneous reaction of $NH_3$ and $NO_2$ (Pathak et al., 2009), and it is considered as a secondary inorganic source (Wei et al., 2024). Factor 4 has high proportions of EC, organic species, $nss–K^+$ and $nss–SO_4^{2-}$. EC and $nss–K^+$ jointly indicate that this factor may have connection with combustion source (biomass burning), while $nss–SO_4^{2-}$ is associated with secondary transformation of $SO_2$ (Dai et al., 2023; Xue et al., 2019). Hence, factor 4 is identified as mixture source (combustion source and $SO_4^{2-}$). Factor 5, with a high proportion of SOC and low proportions of EC and inorganic species, is considered as a secondary organic source. The above proportions are consistent with results of previous studies conducted at BS and YS that have also shown that biomass combustion, secondary organic/inorganic aerosol sources, dust and sea salt are common sources of aerosol components (Zhao et al., 2023; Geng et al., 2020; Zhang et al., 2025).

8.   References: Please edit on the formatting issues in the reference list.

**Author reply:**

We have checked and modified the format of the references.

9.   Supporting Info, Figure S10: The figure resolution is low and the wording is too small. Please revise.

**Author reply:**

For clarity, the original figure has been split into the two separate figures below:

[Figure]

Figure S11. (A) Chromatographic peaks and light absorption chromatogram of $C_{23}H_{31}O_2^-$ and its derivatives. The dotted lines represent the position of the mass spectrometry calibration solution. (B) Secondary mass spectra of $C_{23}H_{31}O_2^-$ (m/z: 339.2339) and its derivatives and their possible structures.

[Figure]

Figure S12. Secondary mass spectrometry of alkylbenzene sulfonic acid and its three fragmentation pathways. The red arrow and molecular formula represent the main fragmentation pathways and products. The red dashed line represents chemical bond breakage.

10.  Supporting Info, Figure S12: What does the dotted line in the figure represent?

**Author reply:**

The dotted line in the Figure represents the position of the mass spectrometry calibration solution. Due to the fact that chromatographic peaks mainly occur in the first 50 minutes of gradient elution, we placed the calibration solution in the last few minutes of gradient elution during sample measurement, to ensure no interference with the analysis of sample chromatographic peaks. However, the mass spectrometry software (Bruker Compass DataAnalysis 4.2) we use cannot remove the two dotted lines. In order to avoid misunderstandings, we have provided explanations for these two dotted lines in the Figure caption. The modified Figure is as follows:

[Figure]

Figure S14. Absorption spectrum of HULIS and WISOC in the TSP and PM$_{2.5}$ samples. The dotted lines represent the position of the mass spectrometry calibration solution.

**References**

Balkanski, Y. J., Jacob, D. J., Gardner, G. M., Graustein, W. C., and Turekian, K. K.: Transport and residence times of tropospheric aerosols inferred from a global three-dimensional simulation of $^{210}$Pb, J. Geophys. Res.-Atmos., 98, 20573–20586, https://doi.org/10.1029/93JD02456, 1993.

Bian, H., Chin, M., Hauglustaine, D. A., Schulz, M., Myhre, G., Bauer, S. E., Lund, M. T., Karydis, V. A., Kucsera, T. L., Pan, X., Pozzer, A., Skeie, R. B., Steenrod, S. D., Sudo, K., Tsigaridis, K., Tsimpidi, A. P., and Tsyro, S. G.: Investigation of global particulate nitrate from the AeroCom phase III experiment, Atmos. Chem. Phys., 17, 12911–12940, https://doi.org/10.5194/acp-17-12911-2017, 2017.

Budhavant, K., Andersson, A., Holmstrand, H., Bikkina, P., Bikkina, S., Satheesh, S. K., and Gustafsson, Ö.: Enhanced Light-Absorption of Black Carbon in Rainwater Compared With Aerosols Over the Northern Indian Ocean, J. Geophys. Res.-Atmos., 125, e2019JD031246, https://doi.org/10.1029/2019jd031246, 2020.

Cohen, M. D., Stunder, B. J. B., Rolph, G. D., Draxler, R. R., Stein, A. F., and Ngan, F.: NOAA's HYSPLIT Atmospheric Transport and Dispersion Modeling System, Bull. Amer. Meteorol. Soc., 96, 2059–2077, https://doi.org/10.1175/bams-d-14-00110.1, 2015.

Dai, Q., Chen, J., Wang, X., Dai, T., Tian, Y., Bi, X., Shi, G., Wu, J., Liu, B., Zhang, Y., Yan, B., Kinney, P. L., Feng, Y., and Hopke, P. K.: Trends of source apportioned PM$_{2.5}$ in Tianjin over 2013–2019: Impacts of Clean Air Actions, Environ. Pollut., 325, 121344, https://doi.org/10.1016/j.envpol.2023.121344, 2023.

Gao, C. Y., Heald, C. L., Katich, J. M., Luo, G., and Yu, F.: Remote Aerosol Simulated During the Atmospheric Tomography (ATom) Campaign and Implications for Aerosol Lifetime, J. Geophys. Res.-Atmos., 127, e2022JD036524, https://doi.org/10.1029/2022jd036524, 2022a.

Geng, X., Mo, Y., Li, J., Zhong, G., Tang, J., Jiang, H., Ding, X., Malik, R. N., and Zhang, G.: Source apportionment of water-soluble brown carbon in aerosols over the northern South China Sea: Influence from land outflow, SOA formation and marine emission, Atmos. Environ., 229, 117484, https://doi.org/10.1016/j.atmosenv.2020.117484, 2020.

Giorgi, F. and Chameides, W. L.: Rainout lifetimes of highly soluble aerosols and gases as inferred from simulations with a general circulation model, J. Geophys. Res.-Atmos., 91, 14367–14376, https://doi.org/10.1029/JD091iD13p14367, 1986.

Huang, S., Wu, Z., Wang, Y., Poulain, L., Höpner, F., Merkel, M., Herrmann, H., and Wiedensohler, A.: Aerosol Hygroscopicity and its Link to Chemical Composition in a Remote Marine Environment Based on Three Transatlantic Measurements, Environ. Sci. Technol., 56, 9613–9622, https://doi.org/10.1021/acs.est.2c00785, 2022.

Kim, G., Cho, H.-j., Seo, A., Kim, D., Gim, Y., Lee, B. Y., Yoon, Y. J., and Park, K.: Comparison of Hygroscopicity, Volatility, and Mixing State of Submicrometer Particles between Cruises over the Arctic Ocean and the Pacific Ocean, Environ. Sci. Technol., 49, 12024–12035, https://doi.org/10.1021/acs.est.5b01505, 2015.

Li, H., Qin, X., Chen, J., Wang, G., Liu, C., Lu, D., Zheng, H., Song, X., Gao, Q., Xu, J., Zhu, Y., Liu, J., Wang, X., Deng, C., and Huang, K.: Continuous Measurement and Molecular Compositions of Atmospheric Water-Soluble Brown Carbon in the Nearshore Marine Boundary Layer of Northern China: Secondary Formation and Influencing Factors, J. Geophys. Res.-Atmos., 128, e2023JD038565, https://doi.org/10.1029/2023jd038565, 2023.

Liu, C., Li, H., Zheng, H., Wang, G., Qin, X., Chen, J., Zhou, S., Lu, D., Liang, G., Song, X., Duan, Y., Liu, J., Huang, K., and Deng, C.: Ocean Emission Pathway and Secondary Formation Mechanism of Aminiums Over the Chinese Marginal Sea, J. Geophys. Res.-Atmos., 127, e2022JD037805, https://doi.org/10.1029/2022jd037805, 2022.

Liu, X., Easter, R. C., Ghan, S. J., Zaveri, R., Rasch, P., Shi, X., Lamarque, J. F., Gettelman, A., Morrison, H., Vitt, F., Conley, A., Park, S., Neale, R., Hannay, C., Ekman, A. M. L., Hess, P., Mahowald, N., Collins, W., Iacono, M. J., Bretherton, C. S., Flanner, M. G., and Mitchell, D.: Toward a minimal representation of aerosols in climate models: description and evaluation in the Community Atmosphere Model CAM5, Geosci. Model Dev., 5, 709–739, https://doi.org/10.5194/gmd-5-709-2012, 2012.

Mo, Y., Zhong, G., Li, J., Liu, X., Jiang, H., Tang, J., Jiang, B., Liao, Y., Cheng, Z., and Zhang, G.: The Sources, Molecular Compositions, and Light Absorption Properties of Water-Soluble Organic Carbon in Marine Aerosols From South China Sea to the Eastern Indian Ocean, J. Geophys. Res.-Atmos., 127, e2021JD036168, https://doi.org/10.1029/2021jd036168, 2022.

Pai, S. J., Heald, C. L., Pierce, J. R., Farina, S. C., Marais, E. A., Jimenez, J. L., Campuzano-Jost, P., Nault, B. A., Middlebrook, A. M., Coe, H., Shilling, J. E., Bahreini, R., Dingle, J. H., and Vu, K.: An evaluation of global organic aerosol schemes using airborne observations, Atmos. Chem. Phys., 20, 2637–2665, https://doi.org/10.5194/acp-20-2637-2020, 2020.

Park, R. J., Jacob, D. J., Field, B. D., Yantosca, R. M., and Chin, M.: Natural and transboundary pollution influences on sulfate-nitrate-ammonium aerosols in the United States: Implications for policy, J. Geophys. Res.-Atmos., 109, D15204, https://doi.org/10.1029/2003jd004473, 2004.

Pathak, R. K., Wu, W. S., and Wang, T.: Summertime $PM_{2.5}$ ionic species in four major cities of China: nitrate formation in an ammonia-deficient atmosphere, Atmos. Chem. Phys., 9, 1711–1722, https://doi.org/10.5194/acp-9-1711-2009, 2009.

Song, J., Zhao, Y., Zhang, Y., Fu, P., Zheng, L., Yuan, Q., Wang, S., Huang, X., Xu, W., Cao, Z., Gromov, S., and Lai, S.: Influence of biomass burning on atmospheric aerosols over the western South China Sea: Insights from ions, carbonaceous fractions and stable carbon isotope ratios, Environ. Pollut., 242, 1800–1809, https://doi.org/10.1016/j.envpol.2018.07.088, 2018a.

Song, M., Tan, Q., Feng, M., Qu, Y., Liu, X., An, J., and Zhang, Y.: Source Apportionment and Secondary Transformation of Atmospheric Nonmethane Hydrocarbons in Chengdu, Southwest China, J. Geophys. Res.-Atmos., 123, 9741–9763, https://doi.org/10.1029/2018jd028479, 2018b.

Stanimirova, I., Rich, D. Q., Russell, A. G., and Hopke, P. K.: A long-term, dispersion normalized PMF source apportionment of $PM_{2.5}$ in Atlanta from 2005 to 2019, Atmos. Environ., 312, 120027, https://doi.org/10.1016/j.atmosenv.2023.120027, 2023.

Tsigaridis, K., Daskalakis, N., Kanakidou, M., Adams, P. J., Artaxo, P., Bahadur, R., Balkanski, Y., Bauer, S. E., Bellouin, N., Benedetti, A., Bergman, T., Berntsen, T. K., Beukes, J. P., Bian, H., Carslaw, K. S., Chin, M., Curci, G., Diehl, T., Easter, R. C., Ghan, S. J., Gong, S. L., Hodzic, A., Hoyle, C. R., Iversen, T., Jathar, S., Jimenez, J. L., Kaiser, J. W., Kirkevåg, A., Koch, D., Kokkola, H., Lee, Y. H., Lin, G., Liu, X., Luo, G., Ma, X., Mann, G. W., Mihalopoulos, N., Morcrette, J. J., Müller, J. F., Myhre, G., Myriokefalitakis, S., Ng, N. L., O'Donnell, D., Penner, J. E., Pozzoli, L., Pringle, K. J., Russell, L. M., Schulz, M., Sciare, J., Seland, Ø., Shindell, D. T., Sillman, S., Skeie, R. B., Spracklen, D., Stavrakou, T., Steenrod, S. D., Takemura, T., Tiitta,

P., Tilmes, S., Tost, H., van Noije, T., van Zyl, P. G., von Salzen, K., Yu, F., Wang, Z., Wang, Z., Zaveri, R. A., Zhang, H., Zhang, K., Zhang, Q., and Zhang, X.: The AeroCom evaluation and intercomparison of organic aerosol in global models, Atmos. Chem. Phys., 14, 10845–10895, https://doi.org/10.5194/acp-14-10845-2014, 2014.

Wei, Y., Wang, S., Jiang, N., Zhang, D., and Zhang, R.: Study on main sources of aerosol pH and new methods for additional reduction of $PM_{2.5}$ during winter severe pollution: Based on the PMF-GAS model, J. Clean Prod., 471, 143401, https://doi.org/10.1016/j.jclepro.2024.143401, 2024.

Xu, F., Hu, K., Lu, S. H., Guo, S., Wu, Y. C., Xie, Y., Zhang, H. H., and Hu, M.: Enhanced Marine VOC Emissions Driven by Terrestrial Nutrient Inputs and Their Impact on Urban Air Quality in Coastal Regions, Environ. Sci. Technol., 59, 8140–8154, https://doi.org/10.1021/acs.est.4c12655, 2025.

Xue, J., Yu, X., Yuan, Z., Griffith, S. M., Lau, A. K. H., Seinfeld, J. H., and Yu, J. Z.: Efficient control of atmospheric sulfate production based on three formation regimes, Nat. Geosci., 12, 977–982, https://doi.org/10.1038/s41561-019-0485-5, 2019.

Zhang, Y., Wang, Y., Li, S., Yi, Y., Guo, Y., Yu, C., Jiang, Y., Ni, Y., Hu, W., Zhu, J., Qi, J., Shi, J., Yao, X., and Gao, H.: Sources and Optical Properties of Marine Organic Aerosols Under the Influence of Marine Emissions, Asian Dust, and Anthropogenic Pollutants, J. Geophys. Res.-Atmos., 130, e2025JD043472, https://doi.org/10.1029/2025jd043472, 2025.

Zhao, S., Qi, J., and Ding, X.: Characteristics, seasonal variations, and dry deposition fluxes of carbonaceous and water-soluble organic components in atmospheric aerosols over China's marginal seas, Mar. Pollut. Bull., 191, 114940, https://doi.org/10.1016/j.marpolbul.2023.114940, 2023.

Zhao, Z. Y., Zhang, Y. L., Lin, Y. C., Song, W. H., Yu, H. R., Fan, M. Y., Hong, Y. H., Yang, X. Y., Li, H. Y., and Cao, F.: Continental Emissions Influence the Sources and Formation Mechanisms of Marine Nitrate Aerosols in Spring Over the Bohai Sea and Yellow Sea Inferred From Stable Isotopes, J. Geophys. Res.-Atmos., 129, e2023JD040541, https://doi.org/10.1029/2023jd040541, 2024.

Zhou, S., Chen, Y., Wang, F., Bao, Y., Ding, X., and Xu, Z.: Assessing the Intensity of Marine Biogenic Influence on the Lower Atmosphere: An Insight into the Distribution of Marine Biogenic Aerosols over the Eastern China Seas, Environ. Sci. Technol., 57, 12741–12751, https://doi.org/10.1021/acs.est.3c04382, 2023.

Zong, Z., Wang, X., Tian, C., Chen, Y., Qu, L., Ji, L., Zhi, G., Li, J., and Zhang, G.: Source apportionment of $PM_{2.5}$ at a regional background site in North China using PMF linked with radiocarbon analysis: insight into the contribution of biomass burning, Atmos. Chem. Phys., 16, 11249–11265, https://doi.org/10.5194/acp-16-11249-2016, 2016.

---

## Author Comment (AC3)

**Reply to Anonymous Referee #1**

We thank Anonymous Referee #1 for the valuable comments on our manuscript. Here we provide point-to-point responses to the Referees' comments. For clarity, the Referees' comments are marked in black, authors' responses are marked in **blue**, and changes in the manuscript are marked in **red**.

This manuscript presents a comprehensive investigation into the chemical composition and sources of aerosols over the Bohai Sea (BS) and Yellow Sea (YS) during summer. The study effectively combines field observations, advanced analytical techniques (including high-resolution mass spectrometry and stable carbon isotope analysis), and receptor modeling to demonstrate the dominant influence of coastal terrestrial emissions, particularly biomass burning, on the aerosol properties of these marginal seas, even under conditions of increased marine air mass influence. The topic is relevant and the conclusions are well-supported by the data. The manuscript is generally well-structured, and the methods are appropriately described. The findings contribute valuable insights to the understanding of land-sea interactions in polluted coastal environments. I recommend publication after addressing the following minor revisions to enhance clarity and impact.

**Specific Comments:**

1. Line 16: The authors wrote "The characteristics of carbon component ratios in aerosols are similar to those in coastal cities", it is better to provide the exact values.

**Author reply:**

We have added the necessary OC/EC and WSOC/OC ratios in the Abstract. Moreover, the carbon component ratios of coastal cities reported in the literature are presented in Table S2 of the Supplement.

This was updated in the revised manuscript at Page 1, Line 16–18:

The characteristics of carbon component ratios (mean: 5.58–12.11 for OC/EC, 0.48–0.58 for WSOC/OC) in aerosols are similar to those in coastal cities, and the proportion

of non–sea–salt ions (> 80%) is significantly higher than that of sea salt ions.

2. In Figure 1, Panel A the running time of the backward trajectories should provided. Panel B full name of the acronyms should be given in the caption, the X-axis title should be added. Panel D is confusing and misleading; the four components should not be placed together.

**Author reply:**

In order to solve the confusion and ambiguity observed in Figure 1, we have added the full name of acronyms and corresponding axis titles in Panel B, redrawn Panel D, and added the running time of the backward trajectory to the figure caption.

The modified Figure 1 is as follows:

[Figure]

**Figure 1. (A) Trajectories of air masses arriving at the Yellow Sea and Bohai Sea during the sampling period. The simulated air mass transport time is 72 hours. Green, yellow and red points represent TSP, PM₂.₅ samples and fire points, respectively. Fire point information comes from https://firms.modaps.eosdis.nasa.gov/map. The yellow dashed line represents the boundary between the northern and southern sea regions. (B) Retention ratio of air masses over land and**

**ocean, as well as the retention ratio of boundary layer air masses. Rmbl stands for Retention ratio of marine boundary layer air masses, Rtbl stands for Retention ratio of terrestrial boundary layer air masses, Rmam stands for Retention ratio of marine air masses and Rtam stands for Retention ratio of terrestrial air masses. (C) Differences in HULIS/WSOC ratio at different sampling points. (D) Proportion of carbonaceous species and water–soluble ions in particles. The pie charts represent the proportion of sea salt ions and non–sea salt ions in the total ions of each sea region. N and S indicate the northern and southern sea regions, respectively.**

3. In Figure 2, The gray circle should change to "The gray circular surface in Panel A".

**Author reply:**

We have modified the description of the assumed boundary layer in the caption of Figure 2.

The gray circular surface in Panel A (at an altitude of 1000 m), and brighter trajectory colors indicate the transport of air masses above the boundary layer.

4. In Figure 3, the meaning of C1, C2 and C3 should given in the caption. Besides, what is the X-axis of Panel A.

**Author reply:**

We have modified Figure 3 and added corresponding X-axis of Panel A and the meaning of C1, C2 andC3 to Figure 3. The revised Figure is shown below:

[Figure]

**Figure 3. (A) Proportion of fluorescent components in WSOC and WISOC. C1–WSOC and C2–WSOC are two humic–like (HULIS) components in WSOC, C3–WSOC is the protein-like (PRLIS) component in WSOC. C1–WISOC and C2–WISOC are HULIS and PRLIS components in WISOC, respectively. (B) Proportion of four types of potential light–absorbing organic compounds in different types of particles.**

5. Line 197-198: Did the author consider all chloride ions to be derived from the sea salts

**Author reply:**

In the manuscript, we used $Na^+$ as the sea salt tracer to estimate the proportion of sea salt and non–sea salt ions, rather than $Cl^-$ as a tracer, which is a commonly used method in field observation studies. The reason is that due to the relative stability of $Na^+$ during the discharge from seawater into the atmosphere, it would neither undergo enrichment nor loss. On the contrary, $Cl^-$ is quite unstable in the atmosphere, as it easily reacts with oxidizing agents (such as OH, $NO_3$, $O_3$) and acidic substances (organic and inorganic acids) to form active chlorine species (Su et al., 2022). This process can change the characteristics of $Cl^-$ ratio to other ions in sea salt. Hence, $Na^+$ is more suitable for

calculating the proportion of sea salt.

In the marginal sea environment, $Na^+$ may not completely come from sea salt due to strong anthropogenic transport. Hence, we use an anthropogenic $Na^+$ to total $Na^+$ ratio range of 50%–80% reported in the literature to estimate the sea salt contribution (as shown in the first paragraph of section 3.2 and in Table S3 in the Supplement) (Wu et al., 2024). No matter how the proportion of $Na^+$ coming from marine sources changes, the results show that non–sea salt ions contribute significantly more to the total ions than sea salt ions. Briefly, $Na^+$ is more representative of sea salt than $Cl^-$, and in this study, changes in the proportion of $Na^+$ from sea salt did not have a disruptive effect on the results.

Table S3. Proportion of sea salt and non-sea salt ions calculated based on different $Na^+$ proportions from the ocean.

| Proportion of $Na^+$ from the ocean | | Sea salt proportion | Non-Sea salt proportion |
|---|---|---|---|
| 100 % $Na^+$ | TSP | 0.16±0.14 | 0.84±0.14 |
| | $PM_{2.5}$ | 0.04±0.03 | 0.96±0.03 |
| 50 % $Na^+$ | TSP | 0.07±0.06 | 0.93±0.06 |
| | $PM_{2.5}$ | 0.02±0.02 | 0.98±0.02 |
| 20 % $Na^+$ | TSP | 0.04±0.03 | 0.96±0.03 |
| | $PM_{2.5}$ | 0.01±0.01 | 0.99±0.01 |

6. Line 202-205: the authors mentioned that "aerosols over the northern sea region exhibited higher OC/EC and lower EC concentration (Fig. S2C), indicating a greater influence of atmospheric transformation on aerosols over the northern sea region. Contrary to the OC/EC, the WSOC/OC was higher in the southern sea region. WSOC/OC is often used as an indicator of secondary organic aerosol (SOA) formation, but this ratio is also influenced by biomass burning or environmental factors". My understanding of this part is that in the norther sea the higher OC/EC ratios indicates secondary organic aerosol formation, the WSOC/OC ratios is higher in the southern sea also indicate secondary organic aerosol formation. If so, why the ratio is contrary between norther and southern sea?

**Author reply:**

Indeed, secondary organic carbon (SOC) formed from photochemical reactions, homogeneous and heterogeneous phases reactions will enhance the concentration of OC and WSOC (Zhang et al., 2023). Hence, high OC/EC ratios and WSOC/OC ratios were commonly considered as the indicators of secondary organic aerosol (SOA) generation despite the uncertainty associated with their characterization using the above method.

Firstly, the characteristics of OC/EC ratio vary among different emission sources, and combustion sources could directly produce WSOC. For example, the OC/EC ratios from vehicle sources (~1.0) are usually lower than those from biomass burning (7.3) (Ram and Sarin, 2010; Cao et al., 2005). In this study, we also found that WSOC correlates well with nss–$K^+$ and EC (biomass burning tracer) (Figure R1A and B). Therefore, in addition to secondary formation, OC and WSOC may be affected by the primary combustion source.

Secondly, organic compounds that cannot be extracted by water may also come from secondary formation. Recent studies have shown that water–soluble organic aerosols (WSOA) account for nearly 59% of organic aerosols (OA) (Zhang et al., 2022b). For example, in this study, we found that some relatively saturated aliphatic compounds were dominated by molecules with O/N ≥ 3, likely suggesting secondary products from reactions of hydrocarbons with $NO_x$ (see last paragraph of Section 3.3), despite their low water solubility. Besides, the SOC calculated using the minimum $R^2$ method has moderate correlation with WSOC and a slope lower than 1 (Figure R1C), further indicating that partial SOC may be water insoluble. Therefore, high OC/EC ratios may not necessarily correspond directly to high WSOC/OC ratios.

Thirdly, OC from non–combustible sources (such as soil or dust) that cannot be extracted by water or organic solvent may also lead to a higher OC/EC ratio. For example, incomplete combustion of biomass can produce plant fibers rich in hydrocarbons, and soil or dust also contribute to carbonaceous species (Gao et al., 2022b; Arun et al., 2021). We found that OC have a strong correlation with nss–$Ca^{2+}$ (dust tracer).

In order to further elucidate that high OC/EC ratios may not correspond to high

WSOC/OC ratios, we summarized the ratio characteristics reported in the literature, and they are shown in Figure R1C below.

[Figure]

**Figure R1. Correlation map of major components in (A) TSP and (B) PM₂.₅. (C) OC/EC and WSOC/OC ratios reported in field studies and the correlation between WSOC and SOC. (D) Concentration comparison of chemical composition in different sea regions.**

We found that the majority of OC/EC and WSOC/OC ratios in field studies are between 5–10 and 0.4–0.7, respectively (Nayak et al., 2022; Arun et al., 2021; Wu et al., 2019; Zhang et al., 2022a; Chen et al., 2020; Chen et al., 2023; Luo et al., 2020; Rajeev et al., 2022; Zhao et al., 2023), in consistency with our results. In addition, there are no similar changes in the trend of OC/EC and WSOC/OC ratios, which means that the generation of SOA may not lead to a simultaneous increase in OC/EC ratios and WSOC/OC ratios. Finally, we believe that the opposite ratio characteristics of OC/EC and WSOC/OC between two sea regions may indicate differences in the contributions from different sources, or different atmospheric transformation characteristics in the two sea regions. Therefore, the sole consideration of the impact of the SOA generation on their ratios

would not be appropriate.

Related to this, we have made revisions in the second paragraph of Section 3.2 as follows (Page 9, Line 217–232):

Regionally, aerosols over the northern sea region exhibited high OC/EC (see Table S2). Contrary to OC/EC, the WSOC/OC ratio was high in the southern sea region. Although higher OC/EC and WSOC/OC may indicate the influence of secondary organic aerosol (SOA), the contrary ratio characteristics in different sea regions indicate that SOA formation is not enough to explain this phenomenon. For example, we found that OC exhibit strong correlations with nss–$Ca^{2+}$ and EC, while WSOC exhibits not only a moderate correlation with secondary inorganic ions ($NH_4^+$, $NO_3^-$ and nss–$SO_4^{2-}$), but also a positive correlation with nss–$K^+$ and EC (Fig. S3A and B). This indicates that in addition to secondary transformation, primary combustion and non–combustion sources contribute to OC and WSOC (Cai et al., 2020). Besides, the SOC calculated using the minimum $R^2$ method exhibits a moderate correlation with WSOC, with a slope lower than 1 (Fig. S3C). This indicates that SOC may partially be water insoluble (Zhang et al., 2022b). The majority of OC/EC and WSOC/OC ratios in field studies fall between 5–10 and 0.4–0.7, respectively (Nayak et al., 2022; Arun et al., 2021; Wu et al., 2019; Zhang et al., 2022a; Chen et al., 2020; Chen et al., 2023; Luo et al., 2020; Rajeev et al., 2022; Zhao et al., 2023) (see Fig. S3C), consistent with the results of this study. In addition, no similar changes were observed between the trends of OC/EC ratios and WSOC/OC ratios, indicating that the generation of SOA may not lead to a simultaneous increase in OC/EC ratios and WSOC/OC ratios. Therefore, combustion sources, non–combustion sources, and atmospheric transformation may be potential reasons for the difference of ratios between the two sea regions. But the EC concentration in the southern sea region is high, indicating that the impact of the primary combustion source seems to be greater (Fig. S3D).

7. Line 205-210: the authors said that "WSOC originates not only from primary biomass burning emissions but also from SOA formation", the authors need to provide more direct evidence. The authors provide WSOC showed significant correlations with

EC and nss–K$^+$, and the spatial distribution of the WSOC/OC aligned with that of EC, nss–K$^+$ and K$^+$ concentration which only indicates the primary sources.

**Author reply:**

We thank the Referee for pointing out this. Indeed, WSOC exhibit a moderate correlation with secondary inorganic ions and SOC as shown in Figure. S3 in Supplement. Necessary revisions have been made in the second paragraph of Section 3.2 in the manuscript.

Page 9, Line 220–223:

For example, we found that OC exhibits strong correlations with nss–Ca$^{2+}$ and EC, while WSOC exhibits not only a moderate correlation with secondary inorganic ions (NH$_4^+$, NO$_3^-$ and nss–SO$_4^{2-}$), but also a positive correlation with nss–K$^+$ and EC (Fig. S3A and B). This indicates that in addition to secondary transformation, primary combustion and non–combustion sources contribute to OC and WSOC (Cai et al., 2020).

[Figure]

**Figure S3. Correlation map of major components in (A) TSP and (B) PM$_{2.5}$. (C) OC/EC and WSOC/OC ratios reported in field studies and the correlation between WSOC and SOC. (D) Concentration comparison of the chemical composition in different sea regions.**

**References**

Arun, B. S., Gogoi, M. M., Hegde, P., Borgohain, A., Boreddy, S. K. R., Kundu, S. S., and Babu, S. S.: Carbonaceous Aerosols over Lachung in the Eastern Himalayas: Primary Sources and Secondary Formation of Organic Aerosols in a Remote High-Altitude Environment, ACS Earth Space Chem., 5, 2493–2506, https://doi.org/10.1021/acsearthspacechem.1c00190, 2021.

Cao, J., Wu, F., Chow, J., Lee, S., Li, Y., Chen, S., An, Z., Fung, K., Watson, J., Zhu, C. J. A. C., and Physics: Characterization and source apportionment of atmospheric organic and elemental carbon during fall and winter of 2003 in Xi'an, China, Atmos. Chem. Phys., 5, 3127–3137, https://doi.org/10.5194/acp-5-3127-2005, 2005.

Chen, H., Yan, C., Fu, Q., Wang, X., Tang, J., Jiang, B., Sun, H., Luan, T., Yang, Q., Zhao, Q., Li, J., Zhang, G., Zheng, M., Zhou, X., Chen, B., Du, L., Zhou, R., Zhou, T., and Xue, L.: Optical properties and molecular composition of wintertime atmospheric water-soluble organic carbon in different coastal cities of eastern China, Sci. Total Environ., 892, 164702, https://doi.org/10.1016/j.scitotenv.2023.164702, 2023.

Chen, P., Kang, S., Tripathee, L., Ram, K., Rupakheti, M., Panday, A. K., Zhang, Q., Guo, J., Wang, X., Pu, T., and Li, C.: Light absorption properties of elemental carbon (EC) and water-soluble brown carbon (WS-BrC) in the Kathmandu Valley, Nepal: A 5-year study, Environ. Pollut., 261, 114239, https://doi.org/10.1016/j.envpol.2020.114239, 2020.

Gao, P., Zhou, C., Lian, C., Wang, X., and Wang, W.: Morphology and Composition of Insoluble Brown Carbon from Biomass Burning, ACS Earth Space Chem., 6, 1574–1580, https://doi.org/10.1021/acsearthspacechem.2c00064, 2022b.

Luo, Y., Zhou, X., Zhang, J., Xue, L., Chen, T., Zheng, P., Sun, J., Yan, X., Han, G., and Wang, W.: Characteristics of airborne water-soluble organic carbon (WSOC) at a background site of the North China Plain, Atmos. Res., 231, 104668, https://doi.org/10.1016/j.atmosres.2019.104668, 2020.

Nayak, G., Kumar, A., Bikkina, S., Tiwari, S., Sheteye, S. S., and Sudheer, A. K.: Carbonaceous aerosols and their light absorption properties over the Bay of Bengal during continental outflow, Environ Sci Process Impacts, 24, 72–88, https://doi.org/10.1039/d1em00347j, 2022.

Rajeev, P., Choudhary, V., Chakraborty, A., Singh, G. K., and Gupta, T.: Light absorption potential of water-soluble organic aerosols in the two polluted urban locations in the central Indo-Gangetic Plain, Environ. Pollut., 314, 120228, https://doi.org/10.1016/j.envpol.2022.120228, 2022.

Ram, K. and Sarin, M. M.: Spatio-temporal variability in atmospheric abundances of EC, OC and WSOC over Northern India, J. Aerosol. Sci., 41, 88–98, https://doi.org/10.1016/j.jaerosci.2009.11.004, 2010.

Su, B., Wang, T., Zhang, G., Liang, Y., Lv, C., Hu, Y., Li, L., Zhou, Z., Wang, X., and Bi, X.: A review of atmospheric aging of sea spray aerosols: Potential factors affecting chloride depletion, Atmos. Environ., 290, 119365, https://doi.org/10.1016/j.atmosenv.2022.119365, 2022.

Wu, G., Ram, K., Fu, P., Wang, W., Zhang, Y., Liu, X., Stone, E. A., Pradhan, B. B., Dangol, P. M., Panday, A. K., Wan, X., Bai, Z., Kang, S., Zhang, Q., and Cong, Z.: Water-Soluble Brown Carbon in Atmospheric Aerosols from Godavari (Nepal), a Regional Representative of South Asia, Environ. Sci. Technol., 53, 3471–3479, https://doi.org/10.1021/acs.est.9b00596, 2019.

Wu, X., Kong, Q., Lan, Y., Sng, J., and Yu, L. E.: Refined Sea Salt Markers for Coastal Cities Facilitating Quantification of Aerosol Aging and PM$_{2.5}$ Apportionment, Environ. Sci. Technol.,

58, 8432–8443, https://doi.org/10.1021/acs.est.3c10142, 2024.

Zhang, J., Qi, A., Wang, Q., Huang, Q., Yao, S., Li, J., Yu, H., and Yang, L.: Characteristics of water-soluble organic carbon (WSOC) in PM$_{2.5}$ in inland and coastal cities, China, Atmos. Pollut. Res., 13, 101447, https://doi.org/10.1016/j.apr.2022.101447, 2022a.

Zhang, X., Zhang, X., Talifu, D., Ding, X., Wang, X., Abulizi, A., Zhao, Q., and Liu, B.: Secondary formation and influencing factors of WSOC in PM$_{2.5}$ over Urumqi, NW China, Atmos. Environ., 293, 119450, https://doi.org/10.1016/j.atmosenv.2022.119450, 2023.

Zhang, Z., Sun, Y., Chen, C., You, B., Du, A., Xu, W., Li, Y., Li, Z., Lei, L., Zhou, W., Sun, J., Qiu, Y., Wei, L., Fu, P., and Wang, Z.: Sources and processes of water-soluble and water-insoluble organic aerosol in cold season in Beijing, China, Atmos. Chem. Phys., 22, 10409–10423, https://doi.org/10.5194/acp-22-10409-2022, 2022b.

Zhao, S., Qi, J., and Ding, X.: Characteristics, seasonal variations, and dry deposition fluxes of carbonaceous and water-soluble organic components in atmospheric aerosols over China's marginal seas, Mar. Pollut. Bull., 191, 114940, https://doi.org/10.1016/j.marpolbul.2023.114940, 2023.

---

## Author Comment (AC4)

**Reply to Anonymous Referee #2**

We thank Anonymous Referee #2 for the valuable comments on our manuscript. Here we provide point-to-point responses to the Referees' comments. For clarity, the Referees' comments are marked in black, authors' responses are marked in **blue**, and changes in the manuscript are marked in **red**.

This study by Hu et al. presents a comprehensive study on the chemical composition, optical properties, and sources of aerosols over the Bohai Sea and Yellow Sea during summer 2023, based on shipboard observations of TSP and PM2.5. The authors combine detailed chemical analyses, optical measurements, stable isotopes, PMF source apportionment, and air mass trajectory analysis to demonstrate that, despite a higher proportion of marine air masses in summer, terrestrial (particularly coastal) emissions dominate the aerosol characteristics in these marginal seas. The topic is important for understanding land–sea interactions in heavily anthropogenically influenced marginal seas. The dataset is valuable and the conclusions are well-supported by the integrated analyses. With some corrections to technical errors and clarifications, this manuscript is suitable for publication in Atmospheric Chemistry and Physics. I recommend minor revision.

Comments:

1. Page 3: Change "from 15 to July 23, 2023," to "from July 15 to 23, 2023". Similarly, change "from 11 to August 13, 2023" to "from August 11 to 13, 2023".

**Author reply:**

This has been corrected. Page 3, Line 75–76:

The cruise campaign was conducted aboard the R/V Lanhai 101, with samples taken in the BS and northern YS from July 15 to 23, 2023, and in the southern YS from August 11 to 13, 2023.

2. Page 3: The flow rate should be 1.05 m$^3$ min$^{-1}$.

**Author reply:**

This has been corrected in the revised manuscript, Page 3, Line 76–78:

Two high–volume air samplers (Laoying 2031) were located on the first deck of the ship to collect TSP and $PM_{2.5}$ samples at a flow rate of 1.05 $m^3$ $min^{-1}$ on quartz filters (preheated at 500°C for 4 hours).

3. Page 3: "a standard solution of 5mg $L^{-1}$ was used…". Specify what ions are in this standard.

**Author reply:**

The standard solution of 5 mg $L^{-1}$ in the manuscript is a mixed standard solution of eight ions including $Na^+$, $NH_4^+$, $K^+$, $Mg^{2+}$, $Ca^{2+}$, $Cl^-$, $SO_4^{2-}$, and $NO_3^-$, with the concentration of each ion being 5 mg $L^{-1}$. This has been updated in the revised manuscript.

Page 3, Line 87–91:

Water–soluble ions ($Na^+$, $NH_4^+$, $K^+$, $Mg^{2+}$, $Ca^{2+}$, $Cl^-$, $SO_4^{2-}$, and $NO_3^-$) were extracted from filters and analyzed by ion chromatography. In order to verify the stability of the instrument, a standard solution of 5 mg $L^{-1}$ of the above-mentioned ions was used to calibrate the instrument after every 5 samples to ensure that the relative standard deviation of repeated measurements of the same sample is less than 6%.

4. Page 7: The choice of 1000 m as the boundary layer height is reasonable, but it can vary due local meteorology. Consider adding sensitivity analysis (e.g., testing 500 and 1500 m) to show if Rtbl/Rmbl values are robust.

**Author reply:**

Based on the Referee's comment, we calculated the values of Rtbl and Rmbl at boundary layer heights of 500 and 1500 m, respectively. The results are shown in following Table R1.

The retention ratio of terrestrial air masses (Rtam) and marine air masses (Rmam) are calculated based on the landing point position during the air mass transport process, and are independent of the boundary layer height. We found that changes in boundary

layer height mainly have impacts on the retention ratio of terrestrial boundary layer air mass (Rtbl) in the northern sea region. When the height of the boundary layer decreases from 1500 m to 500 m, the Rtbl value decreases from $0.63 \pm 0.24$ to $0.34 \pm 0.27$, while the retention ratio of marine boundary layer air mass (Rmbl) remains unchanged. This is mainly attributed to the significant height variation of air masses from northeast China when transported over land, while the variation over the ocean is relatively small. However, no matter how the height of the boundary layer changes, what remains unchanged is that the value of Rmbl is always higher than that of Rtbl. From Figure R1, the height of the boundary layer in the sampling region is mainly below 1000 m or 500 m. Moreover, at boundary layer heights of 1000 m and 500 m, Rtbl changed by about 0.1, without altering its lower characteristics in the northern sea region. Therefore, a boundary layer height of 1000 m can be used as a reference upper limit, and the results are reasonable and robust.

Table R1. Changes in the proportion of four types of air masses in the northern and southern sea regions at different boundary layer heights.

| Boundary layer height (m) | Sea region | Rtam | Rmam | Rtbl | Rmbl |
|---|---|---|---|---|---|
| 1500 | Northern | $0.55 \pm 0.28$ | $0.45 \pm 0.28$ | $0.63 \pm 0.24$ | $0.91 \pm 0.17$ |
| | Southern | $0.31 \pm 0.27$ | $0.69 \pm 0.27$ | $0.60 \pm 0.32$ | $0.99 \pm 0.02$ |
| 1000 | Northern | $0.55 \pm 0.28$ | $0.45 \pm 0.28$ | $0.44 \pm 0.26$ | $0.91 \pm 0.17$ |
| | Southern | $0.31 \pm 0.27$ | $0.69 \pm 0.27$ | $0.60 \pm 0.32$ | $0.99 \pm 0.02$ |
| 500 | Northern | $0.55 \pm 0.28$ | $0.45 \pm 0.28$ | $0.34 \pm 0.27$ | $0.91 \pm 0.17$ |
| | Southern | $0.31 \pm 0.27$ | $0.69 \pm 0.27$ | $0.57 \pm 0.31$ | $0.99 \pm 0.02$ |

[Figure]

Figure R1. Boundary layer height variation of Bohai Sea, Yellow Sea and its surrounding regions (red box) during the sampling period. The boundary layer height data was downloaded from the ERA5 dataset of the European Centre for Medium Range Weather Forecasts (ECMWF): https://cds.climate.copernicus.eu/datasets.

We have added the above Table R1 and Figure R1 to the Supplement, and the following text of Section 3.1 in the revised manuscript has been updated (Page 7, Line 183–186): The satellite observation results also showed that the boundary layer height of the study region during the sampling period was mainly below 1000 m or 500 m, and the difference in retention ratio of air masses calculated at these boundary layer heights was not significant (Table S1 and Fig. S2). Therefore, we chose 1000 m (equivalent to ~900 hPa) as the upper limit of the boundary layer (Deng et al., 2022).

5. Page 8: "…the concentration of particulate matter is significantly correlated with the calculated Rtbl", provide the correlation coefficient here.

**Author reply:**

We have added correlation coefficients and significance levels in the revised manuscript, Page 8, Line 203–205:

Firstly, the concentration of particulate matter is significantly correlated with the calculated Rtbl ($R = 0.829$ and $P < 0.01$ for TSP, and $R = 0.811$ and $P < 0.05$ for $PM_{2.5}$), indicating that aerosols in marginal seas are greatly influenced by terrestrial sources (Fig. S3A and B).

[Figure]

**Figure S3.** Correlation map of major components in (A) TSP and (B) PM$_{2.5}$. (C) OC/EC and WSOC/OC ratios reported in field studies and the correlation between WSOC and SOC. (D) Concentration comparison of the chemical composition in different sea regions.

**References**

Deng, J., Ma, H., Wang, X., Zhong, S., Zhang, Z., Zhu, J., Fan, Y., Hu, W., Wu, L., Li, X., Ren, L., Pavuluri, C. M., Pan, X., Sun, Y., Wang, Z., Kawamura, K., and Fu, P.: Measurement report: Optical properties and sources of water-soluble brown carbon in Tianjin, North China – insights from organic molecular compositions, Atmos. Chem. Phys., 22, 6449–6470, https://doi.org/10.5194/acp-22-6449-2022, 2022.